# Quantum transport theory for unconventional magnets: Interplay of altermagnetism and p-wave magnetism with superconductivity

Tim Kokkeler[1⋆], Ilya Tokatly[2†] and F. Sebastian Bergeret[3‡]

**1** Donostia International Physics Center (DIPC), 20018 Donostia–San Sebastián, Spain and University of Twente, 7522 NB Enschede, The Netherlands
**2** Donostia International Physics Center (DIPC), 20018 Donostia–San Sebastián, Spain, Departamento de Polimeros y Materiales Avanzados, Universidad del Pais Vasco UPV/EHU, 20018 Donostia-San Sebastian, Spain and IKERBASQUE, Basque Foundation for Science, 48009 Bilbao, Spain
**3** Centro de Física de Materiales (CFM-MPC) Centro Mixto CSIC-UPV/EHU, E-20018 Donostia-San Sebastián, Spain and Donostia International Physics Center (DIPC), 20018 Donostia–San Sebastián, Spain

⋆ tim.kokkeler@dipc.org , † ilya.tokatly@ehu.es , ‡ fs.bergeret@csic.es

## Abstract

We present a quantum transport theory for generic magnetic metals, in which magnetism occurs predominantly due to exchange interactions, such as ferromagnets, antiferromagnets, altermagnets and *p*-wave magnets. Our theory is valid both for the normal and the superconducting state and is based on the generalization of nonlinear sigma model to such systems. We derive the effective low-energy action for each of these materials, where the symmetries of the corresponding spin space group are used to determine the form of the tensor coefficients appearing in the action. The transport equations, which are obtained as the saddle point equations of this action, describe a wider range of phenomena than the usual quasiclassical equations. In ferromagnets, in addition to the usual exchange field and spin relaxation effects, we identify a spin-dependent renormalization of the diffusion coefficient, and a correction to the Hanle effect. The former provides a description of spin-polarized currents in both the normal and superconducting equal spin-triplet states. In the normal state, our equations provide a complete description of the spin-splitting effect in diffusive systems, recently predicted in ideal clean altermagnets. In the superconducting state, our equations predict a proximity induced magnetization, the appearance of a spontaneous magnetic moment in hybrid superconductor-altermagnet systems. The distribution and polarization direction of this magnetic moment depend on the symmetry of the structure, thus measurements of such polarization reveal the underlying microscopic symmetry of the altermagnet. Finally, for inversion-symmetry-broken antiferromagnets, such as the *p*-wave magnet, we show that spin-galvanic effects may emerge. This effect is indistinguishable from the spin-galvanic effect induced by spin-orbit coupling in the normal state, but they can be distinguished through the temperature dependence in the superconducting state. Besides these examples, our model applies to arbitrary magnetic systems, providing a complete theory for nonequilibrium transport in diffusive nonconventional magnets at arbitrary temperatures.

# Contents

# 1  Introduction

Transport in junctions that host both superconductivity and magnetism has been a longstanding topic of research [1, 2], which amongst other things has paved the way for the field of superconducting spintronics [3, 4]. The discovery of unconventional types of magnetism such as altermagnetism [5] and $p$ - wave magnetism [6] has opened up new windows within this field, garnering a lot of interest [7–25] because in such materials there is no preferential spin direction and consequently no net exchange field that suppresses superconductivity.

The presence of magnetism considerably affects the proximity effect and therefore transport through hybrid structures with superconductors (SC) and ferromagnets (F). A convenient way to describe transport in hybrid mesoscopic systems is by using quasiclassical theories [26]. Within the framework of quasiclassical theories, ferromagnetism is usually implemented via an exchange field, which accounts for interconversion between the singlet and triplet components of the superconducting condensate. So far, the generalization of the theory to unconventional magnets has only been achieved for a few specific types, namely antiferromagnetism [27–29] and $d$-wave altermagnetism in materials with quadratic dispersion, in the limit of weak superconductivity [30], *i.e.*, when the theory can be linearized with respect to the superconducting order parameter. Moreover, these approaches are developed to describe equilibrium properties, such as supercurrents, and are therefore only valid in the superconducting state. Thus, a transport theory of superconductivity and magnetism, describing the transport in junctions with superconductors and unconventional magnets, ranging from temperatures well below the critical temperature $T_c$ to high temperatures well above $T_c$, is absent.

One of the main hurdles in developing such a theory is that to take into account any momentum dependent splitting of the bands one should go beyond the zeroth order terms in quasiclassical approximation. This significantly complicates the derivations and implies that the theory might depend on the properties of one specific realization of the phenomenon. Generalization of the governing equations for specific cases to slightly other realizations is already a formidable task in approaches based on microscopic theories. An example of this is the derivation of the diffusion equation for superconductors, known as the Usadel equation, with different types of spin-orbit coupling (SOC), both extrinsic [31] and intrinsic [32], which required extremely involved derivations. Recently, however, it was shown that this problem can be circumvented by using a phenomenological approach in which symmetry arguments are employed to determine the terms entering an effective action for diffusive systems, the so-called nonlinear sigma model, allowed in a material up to the lowest orders in the SOC strength. In this way, the Usadel equation was derived straightforwardly for arbitrary types of SOC [33].

In this article, we follow a similar symmetry based approach to determine the diffusive transport equations for materials with generic types of time-reversal symmetry breaking via exchange. To identify the symmetry allowed terms we consider both the general symmetries

of the underlying manifold, which should be obeyed in any material, and material specific symmetries, which are the real space and spin space symmetries of the underlying lattice. There are two main classifications for the latter type of symmetries, magnetic symmetry groups [34, 35], in which rotations in real space and spin space have to come together, and spin space groups [36–47], in which real space rotations and spin rotations can be implemented independently. Thus, the spin space group classification is much broader, every magnetic space group is a spin space group, but not vice versa.

Which of the two should be used depends on whether spin-orbit coupling is significant in the material. Indeed, invariance of an operation under spin-orbit coupling requires rotations in real space to come along with rotations in spin space, and consequently materials in which spin-orbit coupling is important can only be described using magnetic space groups. On the other hand, for materials with magnetic exchange interactions, such as ferromagnets and antiferromagnets, spin-orbit coupling can usually be ignored because it is relativistic, while exchange interactions are not relativistic and thus dominant. Therefore, in the description of materials with exchange, the spin space groups are useful to identify those terms in the action that are allowed without spin-orbit coupling.

These terms dominate the spin dependence of transport, while the additionally allowed terms in the presence of spin-charge coupling are of relativistic magnitude and therefore ignored in the rest of this paper. Using these symmetry characterizations, and the inherent symmetries of the nonlinear sigma model, we provide the quantum transport theory in both normal and superconducting states in materials with general types of spin space groups. Our theory explains both the difference in conductance between different spins in ferromagnets, and transverse effects in altermagnets. It illuminates the consequences of this effect in the superconducting state, and provides the first full theory that can explain transport and nonequilibrium properties in S / unconventional magnet junctions both far below, close to and above the critical temperature.

The paper is structured as follows. In Sec. 2 we discuss the construction of nonlinear sigma models in condensed matter physics and their intrinsic symmetries. Then in Sec. 3, we derive the general action for materials with exchange interactions, Eq. (22) and the corresponding transport equations, Eqs. (23-25). In Sec. 4 we consider which form this action takes for collinear ferromagnets. We show that, in addition to the Larmor precession that follows from the exchange field, our equations, see Eq. (42, 43) show that in the normal state any electrical current through such a system is spin-polarized. We show using Eqs. (53, 54) that in the superconducting state spin-polarized supercurrents arise if equal-spin triplets are created, for example using different domains of ferromagnets. Using this we show that nonreciprocal transport may appear in ferromagnetic structures, something that thus far could not be captured using quasiclassical theories. Next, in Sec. 5 we consider materials without a net exchange field, antiferromagnets and altermagnets. Our results indicate that while in antiferromagnets there is only spin relaxation, in altermagnet, the diffusion constant becomes spin dependent, and next to this, a gradient dependent Hanle effect exists. We show that in hybrid junctions between a superconductor and an altermagnet this leads to a proximity induced magnetization. The sign of the induced magnetization depends on the orientation of the boundary, as summarized in Fig. 2. In Sec. 6 we extend our model to second order to higher order to capture spin-galvanic effects in $p$-wave magnets. We show that this spin-galvanic effect in the normal state is indistinguishable from its spin-orbit induced analog, but in the superconducting state can be distinguished through its temperature dependence, as shown in Fig. 3. We conclude our article with a summary of all effects in Sec. 7.

## 2 Construction of the action

In the coming sections we construct the effective low energy action for different types of magnetic conducting materials in the diffusive regime, when the physics is dominated by the soft modes described by the so-called nonlinear sigma model (NLSM). As explained in the introduction, our derivation is based only on symmetry constraints, regardless of the electronic structure and other microscopic details of specific materials and/or structures. In this way we find all symmetry allowed terms in the action, and then study their physical implications for different materials. For simplicity of notation, we use $\hbar = k_B = c = 1$ throughout the text.

To identify the main rules for the construction of the NLSM in the next sections we first analyze a generic example of a free electronic disorder system with superconducting correlations. Its microscopic action can be compactly written on a contour $\mathcal{C}$ in time-space as:

$$iS = \frac{1}{2}\text{tr}\left(\int_{\mathcal{C}} dt \int d\boldsymbol{r}\,\bar{\boldsymbol{\Psi}}\check{\mathcal{L}}\boldsymbol{\Psi}\right), \tag{1}$$

where tr denotes tracing out the matrix degrees of freedom and the spatial integration is taken over the whole system. For the contour $\mathcal{C}$ in time we choose to use the Keldysh contour, which contains two branches, one that runs from $-\infty$ to $+\infty$ and one that runs in the opposite direction. This choice of contour is suitable for the description of out-of-equilibrium properties [48–51].

In the integrand in Eq. (1), we have introduced the bi-spinors in the so-called Nambu-spin space [51, 52]. These bispinors combine the Grassmann spinors $\psi$ and $\bar{\psi}$ with their time-reversed conjugates:

$$\boldsymbol{\Psi} = \begin{pmatrix} \psi_\uparrow \\ \psi_\downarrow \\ \bar{\psi}_\downarrow \\ -\bar{\psi}_\uparrow \end{pmatrix}, \qquad \bar{\boldsymbol{\Psi}} = \begin{pmatrix} \bar{\psi}_\uparrow, & \bar{\psi}_\downarrow, & -\psi_\downarrow, & \psi_\uparrow \end{pmatrix}. \tag{2}$$

The operator $\check{\mathcal{L}}$ is defined as

$$\check{\mathcal{L}} = i\tau_3\hat{\partial}_t - \tau_3\check{H}, \tag{3}$$

and the Hamiltonian $\check{H}$ has following general structure in the spin-Nambu space, described in this work by the Pauli matrices $\sigma$ and $\tau$ respectively:

$$\check{H} = \check{\xi} - |\Delta|\tau_1 e^{i\tau_3\varphi}. \tag{4}$$

Here, $\check{\xi}$ is the normal state Hamiltonian, which contains both the periodic crystal potential and the disorder potential, while $|\Delta|$ is the magnitude of the superconducting gap. There is a symmetry inherent to this Hamiltonian and the action, Eq. (1). This symmetry stems from the redundancy of the Hilbert space in the Nambu description, which involves the electronic states and its time-reversal counterparts, see Eq. (2). Due to this constriction, the bispinors $\Psi$ and $\bar{\Psi}$ are not independent, they are related via the charge conjugation operation [49,53,54]:

$$\bar{\Psi} = \Psi^T(i\sigma_y\tau_1). \tag{5}$$

Because this is an inherent symmetry of Nambu-space, we may restrict ourselves to operators $\mathcal{O}$ that satisfy the charge conjugation relation, defined by following transformation $\mathcal{O} \to \tau_1\sigma_y\mathcal{O}^T\tau_1\sigma_y$. In particular, this holds true for the microscopic Lagrangian $\hat{\mathcal{L}}$ in Eq. (3) satisfies:

$$\hat{\mathcal{L}} = \tau_1\sigma_y\hat{\mathcal{L}}^T\tau_1\sigma_y. \tag{6}$$

Since charge conjugation is a fundamental symmetry of the constructed space, it must be present at any level of approximation, and we impose it as one of the conditions constraining the form of the effective low energy theory.

For our purpose of describing magnetic systems we also introduce the time-reversal transform for operators:

$$\mathcal{O} \rightarrow \tau_3 \sigma_y \mathcal{O}^T \tau_3 \sigma_y \,. \tag{7}$$

The term in Eq. (4) describing the one particle normal state Hamiltonian, $\hat{\xi}$ can then be decomposed in time-reversal symmetric and antisymmetric parts,

$$\hat{\xi}_s = \frac{1}{2} \left( \hat{\xi} + \sigma_y \hat{\xi}^T \sigma_y \right), \qquad \hat{\xi}_a = \frac{1}{2} \left( \hat{\xi} - \sigma_y \hat{\xi}^T \sigma_y \right), \tag{8}$$

such that the Hamiltonian Eq. (4) can be written as

$$\check{H} = \hat{\xi}_s \tau_3 + \hat{\xi}_a - |\Delta| \tau_1 e^{i \tau_3 \varphi} \,, \tag{9}$$

where $\xi_s$ is invariant under Eq. (7), while $\xi_a$ changes sign under this transformation. It is known that the simultaneous presence of time-reversal and charge conjugation symmetries implies a unitary, so-called chiral symmetry [55]. In the present case, the combination of the operations Eqs. (6) and (7) leads to the chiral symmetry defined by the transformation

$$\mathcal{O} \rightarrow \tau_2 \mathcal{O} \tau_2 \,. \tag{10}$$

We note, that this reflects a generic symmetry of Nambu spinors in time reversal symmetric systems, which implies that the time-reversed partner of $\Psi$ equals to $i\tau_2\Psi$, while the time-reversed partner of $\bar{\Psi}$ is $-i\bar{\Psi}\tau_2$. Terms in the operator $\mathcal{L}$ defined in Eq. (3) that preserve time reversal remain unchanged after chiral transformation (10). Terms breaking time reversal, such as $i\tau_3\hat{\partial}_t$ and $\tau_3\hat{\xi}_a$ change their sign after applying Eq. (10).

In some materials time-reversal is intrinsically broken, like in ferromagnets, via an exchange field. In others time-reversal can be broken via an external magnetic field, or a charge current. Thus, at the level of the Hamiltonian, time-reversal breaking is described in different ways: either via the vector potential $\mathbf{A}$, via exchange or Zeeman terms containing Pauli matrices in the spin space $\boldsymbol{\sigma}$, or through the superconducting phase $\varphi$. In fact, in Eqs. (1, 3, 9) all terms that are antisymmetric with respect to time-reversal, i.e. $\hat{\xi}_a$, the phase $\varphi$ and the vectors $\mathbf{A}$ and $\sigma$, are accompanied by the matrix $\tau_3$ in Nambu space, consistent with the chiral symmetry, Eq. (10). This structure has to be preserved when constructing the NLSM.

The Keldysh NLSM for disordered electron systems, has been derived microscopically in many different situations using different formalisms [31,32,49,51,56–66]. In all of these theories, the rapidly varying and random disorder is taken into account by performing a disorder average. Using this approach, the soft diffusion modes are described in terms of the composite matrix field $Q(\boldsymbol{r}, t, t')$, that is conjugate to the bilinear form $\Psi(\boldsymbol{r}, t)\bar{\Psi}(\boldsymbol{r}, t')$. The generating function $Z = e^{iS}$ can be expressed in these Q-matrices via

$$Z = \int dQ e^{iS[Q]} \,, \tag{11}$$

where $S[Q]$ is the action of the NLSM. For systems in which the Fermi energy is the largest energy scale, and the elastic scattering rate is larger than any other energy, the main contribution to this generating function stems from the soft-mode manifold characterized by the nonlinear constraint $Q^2 = \mathbf{1}$, and it is therefore sufficient to restrict $Q$ to this manifold only.

The NLSM can be used for several types of calculations, including fluctuation calculations and RG-type analyses. Here however, we are mostly interested in one specific property, namely

that, because $Q$ is the auxiliary field of $\Psi\bar{\Psi}$, its expectation value $\langle Q\rangle = \int dQ Q e^{iS[Q]}$ corresponds to the quasiclassical Green's function $g(t,t')$. As usual, since the integral is dominated by those values for which $iS[Q]$ takes an extreme value, we may approximate $g$ by the saddle point configuration of the NLSM. From this we conclude that the saddle point equations determine the quasiclassical Green's function, that is, they can be identified as the quantum kinetic equations, which in the diffusive regime are commonly referred to as the Usadel-type equations.

Our strategy is first to derive all terms allowed to appear in the action of NLSM, and subsequently to find the Usadel equation for $g$ by requiring the stationarity of the action under variations of $Q$ that obey the soft mode manifold constraint.

For the NLSM in Nambu space, by virtue of Eq. (5) we must restrict the soft mode manifold further. Indeed, by construction of $Q$, it inherits the symmetry properties of the bilinear form $\Psi(t)\bar{\Psi}(t')$ [33]. Following Eq. (5), this means

$$Q(t,t') = \tau_1\sigma_y Q^{\mathrm{T}}(t',t)\sigma_y\tau_1\,, \tag{12}$$

where $^{\mathrm{T}}$ is used to denote matrix transposition in Nambu-spin space.

For the calculation of transport properties it is useful to describe the two branches of the Keldysh contour as a matrix degree of freedom, the Keldysh space:

$$Q(t,t') = \begin{bmatrix} Q(t^+, t'^+) & Q(t^+, t'^-) \\ Q(t^-, t'^+) & Q(t^-, t'^-) \end{bmatrix}, \tag{13}$$

where the superscripts refer to the upper (+) and lower (-) branches of the Keldysh contour $\mathcal{C}$. It is customary to perform the following rotation in the space of contour branches $Q \mapsto L\rho_3 Q L^\dagger$, where $L = (1-i\rho_2)/\sqrt{2}$ and $\rho_i$ are the Pauli matrices in the Keldysh space. This transformation ensures that any causal function, such as the Green's function $g$ takes the form

$$g = \begin{bmatrix} g^R & g^K \\ 0 & g^A \end{bmatrix}, \tag{14}$$

where the subscripts $R, A, K$ refer to the retarded, advanced and Keldysh components [49–51, 54]. In this rotated Keldysh space the charge conjugation symmetry reads

$$Q(t,t') = \rho_1\tau_1\sigma_y Q(t',t)^T \sigma_y\tau_1\rho_1\,, \tag{15}$$

where $T$ denotes matrix transposition in Nambu-Keldysh-spin space. Next to this, the NLSM contains one additional symmetry. Because the generator of evolution is Hermitian, the chronological and antichronological propagators are related via Hermitian conjugation. However, in Keldysh space, the chronological and antichronological propagators are also related through a matrix operation in Keldysh space. The equivalence of these two operations imposes a symmetry on the action of the NLSM [33], which we call the chronology symmetry. Within the rotated Keldysh basis, this symmetry requirement reads

$$iS[Q] = (iS[-\rho_2\tau_3 Q^\dagger\tau_3\rho_2])^*\,. \tag{16}$$

To illustrate the terms that appear in an NLSM action that obeys both symmetry requirements in Eqs. (15, 16), we write as an example the well-known action for a diffusive superconductor in the presence of an exchange field [67, 68] and relaxation due to magnetic impurities [69–71]

$$iS_0[Q] = \frac{\pi\nu}{2}\mathrm{Tr}\Big(-\frac{D}{4}(\boldsymbol{\nabla} Q - i[A\tau_3, Q])^2 + \hat{\omega}_{t,t'}\tau_3 Q$$
$$+ \tau_2|\Delta|e^{i\tau_3\varphi} + i\boldsymbol{h}\cdot\boldsymbol{\sigma}\tau_3 Q + \frac{1}{8\tau^s}\tau_3\sigma_a Q\tau_3\sigma_a Q\Big), \tag{17}$$

where Tr denotes tracing over all relevant matrix dimensions and integration over time and space. Here and hereafter, summation over repeated indices is implied. As required, all of these terms obey the charge conjugation symmetry Eq. (15) and chronology symmetry Eq. (16). The action contains the usual kinetic term, the time derivative term $\hat{\omega}_{t,t'} = \delta(t-t')\partial_t$, the pair potential magnitude $|\Delta|$, exchange field $\boldsymbol{h}$ and a spin-relaxation term characterized by the relaxation time $\tau^s$, which is due to scattering on magnetic impurities. The density of states per spin $\nu$ only enters as a prefactor, and therefore it does not enter the transport equations, but instead it appears in the definition of the physical observables.

The action in Eq. (17) describes systems both in and out of equilibrium. Since we describe the system through the generating functional $Z$, in equilibrium, the obtained expressions are naturally related to the free energy $F$ of the system. Indeed, $F$ is obtained by setting $\hat{\omega}_{t,t'} \rightarrow \omega_n$, where $\omega_n = (2n+1)\pi T$ are the Matsubara frequencies and $T$ is temperature, and defining $F = -T\sum_n iS[g, \omega_n]$, where $g$ is the quasiclassical Green's function in the Matsubara representation [72]. This provides a link between the action of NLSM and the Luttinger-Ward free energy functional of the quasiclassical Green function.

The NLSM action for weak ferromagnets in Eq. (17) contains four terms related to time reversal symmetry breaking mechanisms. As mentioned before, all time-reversal symmetry breaking terms, the vector potential, the exchange field, and the superconducting phase, enter the action accompanied by a $\tau_3$, respectively via $\boldsymbol{A}\tau_3$, $\boldsymbol{h}\cdot\boldsymbol{\sigma}\tau_3$ and $\varphi\tau_3$. Next to this, the relaxation term, $1/\tau^s$, is accompanied by two $\tau_3$'s. It originates from a time-reversal symmetry breaking mechanism, exchange of the magnetic impurities, but does not change sign under time-reversal itself because it is of second order.

The exchange field is only nonzero if there is a net magnetization, such as in a ferromagnet or a ferrimagnet, or in a paramagnet that has a Zeeman coupling with an external field. In the normal state this term leads to Larmor precession of a nonequilibrium spin-density [73], whereas in the superconducting state, it is responsible for singlet-triplet conversion [74]. The spin relaxation term is allowed in all materials with magnetic structure on the microscale, and it is even under inversion of all atomic spins. In the normal state it leads to relaxation of electron spin, while in the superconducting state it also leads to a pair breaking effect [75]. We are interested in the transport equation, which is the saddle point equation of the action, Eq. (17). Minimization of the action leads to the well-known Usadel equation for weak ferromagnets [2]. However, as we show below, besides the exchange term, there are more terms allowed in the action of magnetic materials.

In the coming sections we extend the action in Eq. (17) by adding all symmetry allowed terms up to first order in spatial derivatives and $\tau_3\boldsymbol{\sigma}$. We consider metallic systems, in which particle hole symmetry breaking terms are small, and the energy dependence of the coefficients in the action can be ignored.[1] Subsequently we consider which form these terms may take in several different types of materials; ferromagnets, antiferromagnets and altermagnets.

# 3 Action for materials with time-reversal symmetry breaking via exchange

In this work we focus on materials with exchange interactions, such as ferromagnets, antiferromagnets, altermagnets and $p$ - wave magnets. In such materials the spin-dependent fields are nonzero. There are two types of such fields, magnetic type fields, which break time-reversal symmetry, and spin-orbit type fields, which do not break this symmetry. While often both of these types are present, the magnetic type fields usually dominate, because they are nonrelativistic, unlike spin-orbit coupling.

---

[1]For a discussion about the incorporation of energy-dependent terms in the NLSM, see [56].

Therefore, we focus on terms that are due to exchange interactions, that is, those in which $\tau_3$ and spin-Pauli matrices are locked together, and construct all possible scalars using a given amount of $Q$s, spatial derivatives, Pauli matrices and tensors. The number of terms is greatly reduced by the facts that the trace is invariant under cyclic permutation and that the NLSM is defined on the soft-mode manifold $Q^2 = \mathbf{1}$, which allows one to contract multiple $Q$'s, and rearrange the position of spatial derivatives through $\{Q, \partial_k Q\} = 0$. After this, we select the terms that obey the charge conjugation and chronology symmetries. We start by analyzing the simplest possible time-reversal odd scalars that are of the first order in $\tau_3 \boldsymbol{\sigma}$. The first fundamental requirement is that those terms are invariant under charge conjugation, that is, we require them to satisfy $iS[Q] = iS[\rho_1 \tau_1 \sigma_y Q^T \sigma_y \tau_1 \rho_1]$. This restriction imposes restrictions on the tensors of the corresponding terms. In some cases these restrictions can only be met by requiring the tensor to vanish, in which case a term is disallowed. To lowest order, i.e. without spatial derivatives, we may construct one scalar, $h_a \sigma_a \tau_3$, the usual exchange field that already appeared in Eq. (17). To first order in derivatives we may construct two scalars, $\mathrm{Tr}\alpha_{aj}\sigma_a \tau_3 \partial_j Q$ and $\mathrm{Tr}\kappa_{aj}\sigma_a \tau_3 Q \partial_j Q$. While the former, as elaborated in Appendix A, Eqs. (A.6) is allowed by charge conjugation symmetry, the charge conjugation requirement on the second term yields

$$
\begin{aligned}
\mathrm{Tr}\{\kappa_{aj}\sigma_a \tau_3 Q \partial_j Q\} &= \mathrm{Tr}\{\kappa_{aj}\sigma_a \tau_3 (\rho_1 \tau_1 \sigma_y Q^T \sigma_y \tau_1 \rho_1)(\rho_1 \tau_1 \sigma_y \partial_j Q^T \sigma_y \tau_1 \rho_1)\} \\
&= -\mathrm{Tr}\{\kappa_{aj}\sigma_y \sigma_a \tau_3 \sigma_y Q^T \partial_j Q^T\} = \mathrm{Tr}\{\kappa_{aj}\sigma_a^T \tau_3 Q^T \partial_j Q^T\} \\
&= \mathrm{Tr}\{\kappa_{aj}\partial_j Q Q \sigma_a \tau_3\} = -\mathrm{Tr}\{\kappa_{aj}\sigma_a \tau_3 Q \partial_j Q\},
\end{aligned}
\tag{18}
$$

where we used that for any matrix $A$ we have $\mathrm{Tr}(A) = \mathrm{Tr}(A^T)$. This condition is only satisfied if $\kappa_{aj} = 0$, and therefore this term is not allowed to appear in the NLSM action.

Thus, there is only one term of first order in derivatives that is allowed by charge conjugation symmetry, $iS_1 \sim \mathrm{Tr}\{\alpha_{aj}\sigma_a \tau_3 \partial_j Q\}$. This term however is a total derivative, and it can be reduced to the modification of the exchange field at the boundary of the system. Therefore, we disregard this term in our analysis. Up to second order in derivatives, as elaborated in Appendix A.1.3, Eqs. (A.10, A.11) we may write the following contributions allowed by the charge conjugation symmetry under the constraint $Q^2 = \mathbf{1}$:

$$
iS_{2a} = \frac{\pi \nu}{2}\mathrm{Tr}\left(\gamma_{ajk}\tau_3 \sigma_a \partial_j Q \partial_k Q\right),
\tag{19}
$$

$$
iS_{2b} = \frac{\pi \nu}{2}\mathrm{Tr}\left(i\chi_{ajk}\tau_3 \sigma_a Q \partial_j Q \partial_k Q\right),
\tag{20}
$$

where the factor $\frac{\pi \nu}{2}$ was extracted for the uniformity of notation and $\gamma_{ajk}$ and $\chi_{ajk}$ are third rank time-reversal odd tensors with one spin index and two real space indices. By charge conjugation symmetry, they are required to be symmetric in the two real space indices, $\gamma_{ajk} = \gamma_{akj}$ and $\chi_{ajk} = \chi_{akj}$, as derived in Eqs. (A.10, A.11) in Appendix A.1.3. For each of the charge conjugation allowed terms in the NLSM action, we then impose the chronology symmetry requirement in Eq. (16) to determine whether the coefficients of the tensors are real or imaginary. For example, for the coefficient in Eq. (19) we have

$$
\begin{aligned}
\mathrm{Tr}\left(\gamma_{ajk}\sigma_a \tau_3 \partial_j Q \partial_k Q\right) &= \frac{2}{\pi \nu}iS_{2a}[Q] = \frac{2}{\pi \nu}\left(iS_{2a}[-\rho_2 \tau_3 Q^\dagger \tau_3 \rho_2]\right)^* \\
&= \left(\mathrm{Tr}\left(\gamma_{ajk}\sigma_a \tau_3 (-\rho_2 \tau_3 \partial_j Q^\dagger \tau_3 \rho_2)(-\rho_2 \tau_3 \partial_k Q^\dagger \tau_3 \rho_2)\right)\right)^* \\
&= \mathrm{Tr}\left(\gamma_{ajk}^* \partial_k Q \partial_j Q \tau_3 \sigma_a\right) = \mathrm{Tr}\left(\gamma_{ajk}^* \sigma_a \tau_3 \partial_k Q \partial_j Q\right),
\end{aligned}
\tag{21}
$$

where we used that for any matrix $A$, $\left(\mathrm{Tr}(A)\right)^* = \mathrm{Tr}\left(A^\dagger\right)$. Thus, chronology symmetry implies that $\gamma_{ajk} = \gamma_{akj}^*$. By symmetry in the real space indices, this means $\gamma_{ajk}$ is real. The derivation for the other term is similar, and also $\chi_{ajk}$ is real, see Appendix A.1.3, Eq. (A.13) for

details. The allowed components of the tensor depend on the spin space group of the material under consideration. Their magnitude is not restricted by symmetry considerations, but our expansion in $\tau_3\sigma_a$ assumes these coefficients are small compared to the usual diffusion contribution. In the coming sections we discuss the restrictions on these tensors posed by the crystal structure of the materials.

Thus, the general action up to first order in $\tau_3\boldsymbol{\sigma}$ for materials with exchange interactions is

$$
iS_M[Q] = \frac{\pi\nu}{2}\mathrm{Tr}\bigg( -\frac{D_{jk}}{4}\partial_j Q\partial_k Q + \hat{\omega}_{t,t'}\tau_3 Q + \tau_2|\Delta|e^{i\tau_3\varphi}Q + ih_a\sigma_a\tau_3 Q
$$
$$
+ \frac{1}{8}\Gamma_{ab}\tau_3\sigma_a Q\tau_3\sigma_b Q + \gamma_{ajk}\tau_3\sigma_a\partial_j Q\partial_k Q + i\chi_{ajk}\tau_3\sigma_a Q\partial_j Q\partial_k Q\bigg). \quad (22)
$$

where positive definite symmetric tensors $D_{jk}$ and $\Gamma_{ab}$ correspond to the tensors of the diffusion coefficient and the spin relaxation rate, respectively [33]. As usual, in the following we assume $D_{jk} = D\delta_{jk}$ for clarity of presentation. The results can be straightforwardly generalized to incorporate the matrix structure of $D$. This action is valid both in and out of equilibrium, and even with time-dependent drives.

By variation of the action with respect to $Q$, under the constraint $Q^2 = 1$ around the saddle point, the Usadel equation is obtained. This saddle point can be identified with the quasiclassical Green's function $g(t,t')$ of the system. As elaborated in Appendix B, the Usadel equation can be conveniently expressed in terms of the current and the torques as

$$
\partial_k J_k = [g, \hat{\omega}_{t,t'}\tau_3 + i(h_a\sigma_a)\tau_3 + \hat{\Delta}] + \Gamma_{ab}[g, \tau_3\sigma_a g\tau_3\sigma_b] + \mathcal{T}, \quad (23)
$$

where we introduced $\hat{\Delta}$ as shorthand notation for the matrix pair potential The matrix current $J_k$ and torque $\mathcal{T}$ of the system are given by

$$
J_k = -Dg\partial_k g + \frac{1}{4}\gamma_{ajk}\{\tau_3\sigma_a + g\tau_3\sigma_a g, g\partial_j g\} + \frac{i}{4}\chi_{ajk}[\tau_3\sigma_a + g\tau_3\sigma_a g, \partial_j g], \quad (24)
$$

while the torque becomes

$$
\mathcal{T} = \frac{1}{4}\gamma_{ajk}[\tau_3\sigma_a, \partial_j g\partial_k g] + \frac{i}{4}\chi_{ajk}[\tau_3\sigma_a, g\partial_j g\partial_k g]. \quad (25)
$$

As elaborated in Sec. 2, the quasiclassical Green's function $g$ in the rotated Keldysh space has the well known causality structure:

$$
g(t,t') = \begin{bmatrix} g^R(t,t') & g^K(t,t') \\ 0 & g^A(t,t') \end{bmatrix}, \quad (26)
$$

where $g^{R,A}(t,t')$ are the retarded and advance components describing spectral properties of the system, whereas $g^K$ is the Keldysh component, which due to the normalization condition $g^2 = 1$ can be parameterized as [26]:

$$
g^K(t,t') = \int_{-\infty}^{\infty} dt_1 g^R(t,t_1)F(t_1,t') - F(t,t_1)g^A(t_1,t'), \quad (27)
$$

with $F$ being the matrix distribution function. Notice that all matrices $g^{R,A,K}$ and $F$ are 4×4 matrices in Nambu-spin space.

Eqs. (22-25) provide the action and transport equations of general materials with magnetic structures and describe all the effects discussed in this work. In the following sections, we discuss the form the action and Usadel equation take in different materials, specifically, in ferromagnets, antiferromagnets and altermagnets.

## 3.1 Collinearity

To restrict the forms of the tensors $\gamma_{ajk}$ and $\chi_{ajk}$ in Eq. (22) for the different types of magnetic structure, we need to consider the symmetries of the materials under consideration. The first, and most general important distinction we can make is based on the spin-only group [43, 45, 46], which describes the invariance of the crystal under spin space symmetry operations that do not involve real space. We may distinguish three different classes of magnetic structures with different spin only groups, (i) collinear spins, in which case all spins point along a specific axis and the spin-only group contains a rotation axis, (ii) coplanar spins, in which case the spins together span a plane in spin space and the spin-only group contains a mirror operation, (iii) noncoplanar spins, in which case the spins together span the whole spin space and the spin-only group is the trivial group.

To understand the restrictions posed by the spin-only group, we consider a collinear material whose spins are all along the z-axis. Such material is invariant under rotations in spin space around this axis, and specifically under a $\pi$-rotation around this axis. Since this rotation changes the sign of components with $a = x, y$ in $\gamma_{ajk}$ and $\chi_{ajk}$, we may conclude that only the components with $a = z$ can be nonzero, and hence we may write them as the direct product of the unit vector $\hat{z}$ in spin space and a second rank tensor in real space. For generic orientations $P_a$ of the collinear axis we may write

$$\gamma_{ajk} = \frac{D}{4} P_a T_{jk}, \tag{28}$$

$$\chi_{ajk} = \frac{D}{4} P_a K_{jk}, \tag{29}$$

where $P_a$ is a vector parallel to the collinear axis, to be called the magnetic polarization vector, and $T_{jk}$ and $K_{jk}$ are symmetric second rank tensors which are even under time reversal. Since many magnetic structures are collinear, we use this form in the forthcoming discussion of ferromagnets, antiferromagnets and altermagnets.

The specific structure of the tensors $T_{jk}$ and $K_{jk}$ is determined by the spin symmetry group of the material. According to the general concept of the spin space groups [36–47], this structure is independent of the orientation of the collinear axis, but fully determined by the symmetries with respect to rotations and reflections. Depending on whether a specific rotation or reflection in the spin symmetry group is accompanied with the spin flip (time reversal) or not, the material tensors $T_{jk}$ and $K_{jk}$ should be either antisymmetric or symmetric under this operation.

In the coming sections, Sec. 4 and 5, we explore three different collinear magnetic structures, ferromagnets, antiferromagnets and altermagnets. For each of these materials we identify the allowed terms in the action and their physical consequences. We then explore one consequence of noncollinearity through the example of $p$ - wave magnets in Sec. 6.

## 4 Collinear ferromagnets

In this section, we focus on collinear ferromagnets, *i.e.* materials with a nonvanishing net magnetization of a unit cell, pointing along a certain direction. They form the most well-known class of magnetic structures. In such materials, both $\boldsymbol{h}$ and $\boldsymbol{P}$ in Eqs. (28) and (29), are required to be along the collinear axis, while the spin relaxation allows for two independent components describing two different relaxation rates for spins parallel and perpendicular to the local magnetization. The diffusion constant $D_{jk}$ and the tensors $T_{jk}, K_{jk}$ in Eq. (22) in ferromagnets satisfy the same set of symmetries. For illustrative purposes we consider a material with cubic symmetry and choose them to be proportional to the identity matrix, that is,

$T_{jk} = \gamma \delta_{jk}$ and $K_{jk} = \chi \delta_{jk}$. With this, the action in Eq. (22) adapted to ferromagnets becomes

$$iS_F[Q] = \frac{\pi \nu}{2} \text{Tr} \left( -\frac{D}{4}(\nabla Q)^2 + \hat{\omega}_{t,t'} \tau_3 Q + \tau_2 |\Delta| e^{i\tau_3 \varphi} + i\mathbf{h} \cdot \sigma \tau_3 Q \right.$$
$$\left. + \frac{1}{8} \Gamma_{ab} \tau_3 \sigma_a Q \tau_3 \sigma_b Q + \gamma \frac{D}{4} P_a \sigma_a \tau_3 (\nabla Q)^2 + \chi \frac{iD}{4} P_a \sigma_a \tau_3 Q(\nabla Q)^2 \right). \quad (30)$$

Thus, we identify that in collinear magnets, next to the exchange field, there are two other coefficients that can be nonzero, $\gamma$ and $\chi$. They do not appear in the usual Usadel equation stemming from the action in Eq. (17), but since they are symmetry allowed, they are expected to be nonzero in realistic ferromagnets

By variation of the action with respect to $Q$, we find that the Usadel equation reads

$$\partial_k J_k = [g, \hat{\omega}_{t,t'} \tau_3 + ih_a \sigma_a \tau_3 + \hat{\Delta}] + \Gamma_{ab}[g, \tau_3 \sigma_a g \tau_3 \sigma_b] + \mathcal{T}, \quad (31)$$

where the matrix current and torque of the system are given by

$$J_k = -Dg\partial_k g + \frac{D}{4}\gamma P_a \{\tau_3 \sigma_a + g\tau_3 \sigma_a g, g\partial_k g\} + \frac{iD}{4}\chi P_a[\tau_3 \sigma_a + g\tau_3 \sigma_a g, \partial_k g], \quad (32)$$

while the torque in ferromagnets reads

$$\mathcal{T} = \frac{D}{4}\gamma P_a[\tau_3 \sigma_a, (\partial_k g)^2] + \frac{iD}{4}\chi P_a[\tau_3 \sigma_a, g(\partial_k g)^2]. \quad (33)$$

Equations (31-33) describe the diffusive transport in metallic ferromagnets. Although this system has been widely studied in both normal and superconducting states, to the best of our knowledge, a derivation of these equations that is valid in both states and accounts for magnetic polarization has not been previously presented, except in the limit of a strong ferromagnet [76]. Here, we show that this equation naturally arises from the NLSM by considering all symmetry allowed terms up to second order in derivatives and first order in $\sigma$ terms. In the following subsections, we study physical implications of the new terms in the Usadel equation by applying it to describe the spin-polarized transport in both normal and superconducting states.

## 4.1 Diffusion equation for metallic ferromagnets

In this subsection we derive the diffusion equations for collinear ferromagnets from Eqs. (31-33). In the normal state, the retarded and advanced components of $g$, cf. Eqs. (26-27), are given by [77,78]

$$g^R(t,t') = -g^A(t,t') = \tau_3 \delta(t-t'), \quad (34)$$

and consequently, the Keldysh component in Eq. (27), reads

$$g^K(t,t') = \int_{-\infty}^{\infty} dt_1 g^R(t,t_1) F(t_1,t') - F(t,t_1) g^A(t_1,t')$$
$$= 2F(t,t')\tau_3. \quad (35)$$

The matrix distribution function consists, in principle of 8 independent components, four of them related to charge and spin properties (those proportional to $\tau_3$ and $\tau_0 \sigma_{x,y,z}$) and four of them related to thermal and spin thermal properties ($\tau_0$ and $\tau_3 \sigma_{x,y,z}$) [79].

The substitution of Eq. (35) into the Usadel equation, Eq. (31), results into a diffusion equation for the distribution matrix $F$. Here we are interested only on charge and spin transport. Moreover, since $g$ commutes with $\tau_3$, the Usadel equation does not explicitly contain

$t - t'$ and we may focus only on $F(t, t)$, from which we may compute the observables. Next to this, one can easily verify that, in the normal state, the torque in Eq. (33) vanishes due to the trivial form of the retarded and advanced components, Eq. (34). Consequently, the charge and diffusion equations are given by

$$\partial_t \delta n + \partial_k j_k = 0 \,, \tag{36}$$

$$\partial_t \delta S_b + \partial_k j^s_{ka} = -\Gamma_{ab} \delta S_b \,, \tag{37}$$

where the charge accumulation $\delta n$ and spin accumulation $\delta S$ are related to the chemical potential $\mu$, the spin-chemical potential $\mu^s$ and the quasiclassical Green's function $g$ via

$$\delta n = 2\nu\mu = \frac{\pi\nu}{4}\text{tr}\left(g^K(t,t)\right) = \frac{\pi\nu}{2}\text{tr}\left(\tau_3 F(t,t')\right), \tag{38}$$

$$\delta S_b = 2\nu\mu^s_b = \frac{\pi\nu}{4}\text{tr}\left(\tau_3\sigma_b g^K(t,t)\right) = \frac{\pi\nu}{2}\text{tr}\left(\sigma_b F(t,t')\right). \tag{39}$$

Using the general definition of the matrix current Eq. (32) we can relate the charge current,

$$j_k = \frac{\pi\nu}{4}\text{tr}\left(\tau_3 J^K_k(t,t)\right), \tag{40}$$

and the spin current,

$$j^s_{ka} = \frac{\pi\nu}{4}\text{tr}\left(\sigma_a J^K_k(t,t)\right), \tag{41}$$

to the gradients of the chemical and spin-chemical potentials,

$$j_k = -\sigma_D \partial_k \mu - \gamma\sigma_D P_a \partial_k \mu^s_a \,, \tag{42}$$

$$j^s_{ka} = -\sigma_D \partial_k \mu^s_a - \gamma\sigma_D P_a \partial_k \mu + \chi\sigma_D P_b \varepsilon_{abc} \partial_k \mu^s_c \,, \tag{43}$$

where $\sigma_D = 2e^2\nu D$ is the normal state conductivity. The term proportional to $\chi$, appearing only in the spin current expression, Eq. (43), describes a gradient correction to the Larmor precession of a spin in a nonparallel field, and it is usually neglected. Moreover, in the standard quasiclassical approach also $\gamma = 0$. This prevents any distinction between spin-up and spin-down diffusion or conductivity. In spintronics, however, it is customary to describe spin-dependent transport using simple phenomenological diffusion equations in which the conductivities for spin-up and spin-down electrons differ. Our present approach shows that such a distinction naturally arises by including the symmetry-allowed term $\gamma$ in the action Eq. (30). From Eq. (43), one can directly see that a charge current in a ferromagnet induces a spin-polarized current, allowing us to define spin-dependent conductivities in terms of the parameters of our model:

$$\sigma_{\uparrow,\downarrow} = \frac{1}{2}\sigma_D(1 \pm \gamma P), \tag{44}$$

where $P$ is the magnitude of the vector $P_a$. Here $\uparrow(\downarrow)$ refers to up- and down electrons with respect to the magnetization axis of the ferromagnet which is chosen parallel to the z-axis.

Thus, Eqs. (42-43) can be written more customarily:

$$j_k = -\sigma_\uparrow \partial_k \mu_\uparrow - \sigma_\downarrow \partial_k \mu_\downarrow \,, \tag{45}$$

$$j^s_{kz} = -\sigma_\uparrow \partial_k \mu_\uparrow + \sigma_\downarrow \partial_k \mu_\downarrow \,, \tag{46}$$

with $\mu_{\uparrow,\downarrow} = \mu \pm \mu^s_z$.

The above example illustrates how adding second-order derivatives to the action enables the description of spin-dependent transport coefficients in ferromagnets. This result is not surprising, as it can be easily obtained from phenomenological considerations. However, our action is also valid in the superconducting state. As we show in the next section, our model becomes particularly useful in this context, providing new equations capable of describing spin-polarized supercurrents in diffusive systems with magnetic sublattices.

## 4.2 Spin-polarized currents in diffusive S / F systems

Since the prediction of triplet superconducting correlations in ferromagnets proximitized by superconductors [2], the possibility of spin-polarized supercurrents has opened up a new research field called superconducting spintronics [3, 4]. Most experiments in this field are conducted with metallic diffusive hybrid S / F structures, while theoretical descriptions rely on the Usadel equation. However, as in the normal state, the customary quasiclassical approach does not allow for the description of spin polarization: in other words, all predictions of spin-polarized supercurrents could not be described within the standard Usadel equation. This section shows that physically expected spin-polarized supercurrents appear naturally if the spin-dependent second-order derivative terms in the action, Eq. (30) are included.

We consider the superconducting proximity effect in ferromagnets in equilibrium situation. In this case, transport is dissipationless, and instead of using the full Keldysh representation it is enough to consider the retarded component of the Green's functions, see Eq. (26). In the superconducting state the retarded matrix is not any more diagonal in Nambu space, it takes the form [77, 78]

$$g^R = \begin{bmatrix} \hat{g} & \hat{f} \\ \hat{\tilde{f}} & -\hat{g} \end{bmatrix}, \qquad (47)$$

where the real part of $\hat{g}$ describes the density of states of the material and $\hat{f}, \hat{\tilde{f}}$ are the pair amplitudes. The theory developed above is, unlike the Ginzburg-Landau functional, valid for any temperature, both well below and close to the critical temperature. In general, the resulting Usadel equation is highly nonlinear. However, in certain limits, they can be linearized. Close to the critical temperature, or for weakly transparent interfaces, the pair amplitudes are small and one can expand the Green's function in the pair amplitudes, $f$ around its normal state solution $g^R(t, t') \approx \tau_3 \delta(t - t') + f(t, t')$. Here $f$ is a 4×4 matrix in Nambu-spin space given by

$$f = \begin{bmatrix} 0 & \hat{f} \\ \hat{\tilde{f}} & 0 \end{bmatrix}, \qquad (48)$$

and $\hat{f}, \hat{\tilde{f}}$ are 2×2 matrices in spin space. We choose the collinear axis to be along the $z$-direction. In that case it is convenient to write the pair amplitudes following Eq. (2) as

$$\hat{f} = \begin{bmatrix} f_+ & f_{\uparrow\uparrow} \\ f_{\downarrow\downarrow} & f_- \end{bmatrix}, \qquad \hat{\tilde{f}} = \begin{bmatrix} \tilde{f}_+ & \tilde{f}_{\downarrow\downarrow} \\ \tilde{f}_{\uparrow\uparrow} & \tilde{f}_- \end{bmatrix}. \qquad (49)$$

The diagonal elements of this matrix of pair amplitudes can be written as $f_\pm = f_s \pm f_t$, where $f_s$ is the singlet component, and $f_t$ is the triplet with zero spin projection component. Meanwhile, $f_{\uparrow\uparrow}$ and $f_{\downarrow\downarrow}$ correspond to the equal spin pairs with spin projection ±1 along the collinear axis respectively. The components of $\hat{\tilde{f}}$ are defined in a similar way using Eq. (2).

We now substitute $g^R$ into the Usadel equations, Eqs. (31-33) and keep only terms up to linear order in the pair amplitudes. In equilibrium we may go to the Matsubara representation by defining the Matsubara Green function as $g(\omega_n) = g^R(i\omega_n)$, which corresponds to the replacement $\hat{\omega}_{t,t'} \to \omega_n$ in the Usadel equation, where $\omega_n = (2n + 1)\pi T$ is the Matsubara frequency. In this representation, the approximate Green function takes the form $g(\omega_n) \approx \tau_3 \text{sign}(\omega_n) + f(\omega_n)$, and we obtain the following set of linearized Usadel equations for the pair amplitudes,

$$D(1 \pm iP\chi \text{sign}(\omega_n))\partial_{jj} f_\pm = 2(|\omega_n| \pm ih\text{sign}(\omega_n))f_\pm, \qquad (50)$$

$$D(1 + \gamma P)\partial_{jj} f_{\uparrow\uparrow} = 2|\omega_n| f_{\uparrow\uparrow}, \qquad (51)$$

$$D(1 - \gamma P)\partial_{jj} f_{\downarrow\downarrow} = 2|\omega_n| f_{\downarrow\downarrow}, \qquad (52)$$

and an identical set for the components of $\hat{\tilde{f}}$. Firstly, from Eq. (50), it becomes clear that $f_\pm$, and hence the singlet and zero-spin triplet components, are suppressed by the exchange field and are therefore commonly referred to as short-range. In contrast, equal-spin triplets, which are unaffected by the exchange field, are usually denoted as long-range. As required by charge conjugation symmetry, Eq. (15), the induced triplet pair amplitudes are odd-frequency, consistent with previous works on triplet superconducting condensate in dirty materials [2,80].

Also note that the term proportional to $\chi$ in the equations above, simply renormalizes the exchange field and mixes singlets with short-range triplets. Usually, this term is a small correction to the exchange itself and can be safely neglected, as we did in the normal state.

More interesting are the terms proportional to $\gamma$ in the above equations. These terms are responsible for the spin-dependent conductivities in the normal state, as discussed in the previous section. Similarly, in the superconducting state, the $\gamma$ term leads to different diffusion constants for up- and down-spin polarized pair amplitudes, while leaving the singlet and short-range triplet components unaltered.

To illustrate this we consider a heterostructure including superconductors and ferromagnets, for example, an S / F / S Josephson junction, and compute the charge and spin supercurrents in the F region. These are obtained from Eq. (32) by taking the following traces, $e^2 \frac{\pi\nu}{2}\mathrm{tr}(\rho_1\tau_3 J_k)$ and $e^2 \frac{\pi\nu}{2}\mathrm{tr}(\rho_1\sigma_z J_k)$, *cf.* Eqs. (40-41). Because we are focusing here on equilibrium properties, we use the equilibrium distribution functions to convert the integration over energies to a sum over Matsubara frequencies. The leading terms are second order in the pair amplitudes:

$$I = 2\pi T \sum_n \left( \sigma_\uparrow f_{\uparrow\uparrow}\partial_x \tilde{f}_{\uparrow\uparrow} + \sigma_\downarrow f_{\downarrow\downarrow}\partial_x \tilde{f}_{\downarrow\downarrow} + \frac{\sigma_D}{2}(f_+\partial_x\tilde{f}_+ + f_-\partial_x\tilde{f}_-) - (f \leftrightarrow \tilde{f}) \right), \qquad (53)$$

$$I^s = 2\pi T \sum_n \left( \sigma_\uparrow f_{\uparrow\uparrow}\partial_x \tilde{f}_{\uparrow\uparrow} - \sigma_\downarrow f_{\downarrow\downarrow}\partial_x \tilde{f}_{\downarrow\downarrow} + \frac{\sigma_D}{2}(f_+\partial_x\tilde{f}_+ - f_-\partial_x\tilde{f}_-) - (f \leftrightarrow \tilde{f}) \right). \qquad (54)$$

The second lines of each of these expressions describe the contribution to the current from the singlet and short-range triplet components. Thus, if the F region in the S / F / S junction is long enough, they can be neglected. The first lines describe the contributions from the equal-spin pair amplitudes. Finite polarization of the supercurrent is only possible if $\gamma$ is nonzero, or equivalently if $\sigma_\uparrow \neq \sigma_\downarrow$. As discussed in [74,81], such long-range equal spin triplet currents can be created by using a ferromagnetic region with noncolinear domains.

The above example illustrates how to describe spin-polarized supercurrents and is the first demonstration of this effect within the quasiclassical Usadel formalism.

Another feature that has not been captured using the standard quasiclassical formalism without spin-charge coupling, is the appearance of anomalous currents, which are the supercurrents flowing even in the case of a uniform phase. Such effects may appear in the presence of an exchange field in materials with broken inversion symmetry. Apparently, these conditions are met in a setup with different types of ferromagnetic domains.

However, in a material with only exchange correlations, the anomalous supercurrents are proved to be absent when described at the level of the theory defined by Eq. (17) [82]. The reason is the so-called quasiclassical symmetry [82] of Green functions that follow from Eq. (17):

$$g(\omega_n, h) = \sigma_y\tau_1 g^*(-\omega_n, -h)\sigma_y\tau_1. \qquad (55)$$

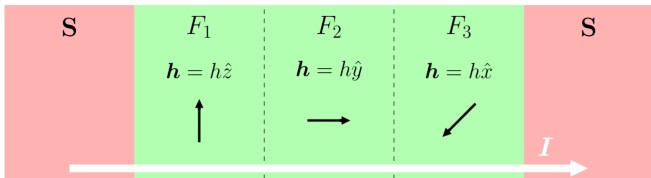

Figure 1: A Josephson junction in which the weak link consists of three different ferromagnets with mutually orthogonal orientations supports an anomalous current $I$ carried by long-range triplets that is proportional to the parameter $\gamma$ that governs the difference in diffusion constant for opposite spins.

This special symmetry of the standard quasiclassical theory forbids the existence of a current, which appears to be even in $h$ through,

$$
\begin{aligned}
j(h) &= \text{Tr} \sum_{\omega} g(\omega, h) \partial_k g(\omega, h) \\
&= \text{Tr} \sum_{\omega} \sigma_y \tau_1 g^*(-\omega, -h) \tau_1 \sigma_y \sigma_y \tau_1 \partial_k g^*(-\omega, -h) \tau_1 \sigma_y \\
&= \left( \text{Tr} \sum_{\omega} (g(-\omega, -h)) \partial_k g(-\omega, -h) \right)^* = j(-h)^* \\
&= j(-h).
\end{aligned}
\tag{56}
$$

This relation, when combined with the with the time-reversal symmetry $j(h) = -j(-h)$, leads to the vanishing anomalous currents at this level of the theory.

We note that the vanishing of the anomalous current in the usual Usadel equation is due to a symmetry that is not a symmetry of the material, and instead of the theoretical framework. Thus, by expanding the theoretical framework, a description of anomalous currents is possible. Until now, describing purely magnetic anomalous currents within the Usadel formalism has only been possible by assuming effective boundary conditions between the superconductor and the ferromagnet, incorporating spin-filtering [82]. Within our theory, which includes the spin-dependent second order derivative terms, we note that while the $\chi$-term obeys Eq. (55), the $\gamma$-term violates the quasiclassical symmetry. It naturally polarizes the pair amplitudes, and therefore our generalized Usadel equations are capable of describing anomalous currents in the presence of long-range triplets. Indeed, by solving Eq. (31) for $S/F_1/F_2/F_3/S$ structure with the F region consisting of three domains with three different magnetizations $\mathbf{m}_{1,2,3}$, illustrated in Fig. 1, one can demonstrate that the magnitude of the anomalous current due to the $\varphi_0$-effect is proportional to satisfies

$$
j_x \sim \gamma \boldsymbol{m_3} \cdot (\boldsymbol{m_1} \times \boldsymbol{m_2}).
\tag{57}
$$

We refer to Appendix C for a derivation of this expression.

## 5 Altermagnets and antiferromagnets

So far, we have focused on ferromagnets, in which there is a net exchange field after averaging over the unit cell. This however need not be the case. There exist magnetic materials in which the magnetization at different sites inside the unit cell is locally nonzero, but it averages to zero yielding a vanishing net exchange field. Examples of such materials, in which the unit cell magnetization is identically zero for symmetry reasons, are antiferromagnets and altermagnets.

Since the nonlinear sigma model describes fields on mesoscopic scales, much larger than the size of the unit cell, its coefficients appear as effectively averaged over the microscopic scales due to the coarse-graining in the formulation of the low-energy theory. Similarly to the macroscopic transport coefficients, the coefficients in NLSM reflect only the macroscopic crystal symmetry. In particular, in materials with zero unit cell magnetization, no effective exchange field should appear in NLSM, that is, $h_a = 0$. The spin-relaxation term is allowed, because it is of second order in $\tau_3 \sigma$. Moreover, the terms containing the tensors $T_{jk}$ and $K_{jk}$ that determine the spin-dependent gradient terms in Eq. (58) are also in general allowed independent of presence of the net magnetization. The specific form of $T_{jk}$ and $K_{jk}$ can be quite different for different classes of magnetic materials, and it is determined by the specific spin space group, as discussed below.

Taking into account all these considerations, the effective action for a diffusive conductor with vanishing magnetization of the unit cell, obtained from Eq. (22), takes the form:

$$iS_A[Q] = \frac{\pi \nu}{2} \text{Tr} \bigg( -\frac{D}{4}(\nabla Q)^2 + \hat{\omega}_{t,t'}\tau_3 Q + \tau_2 |\Delta| e^{i\tau_3 \varphi} + \frac{1}{8}\Gamma_{ab}\tau_3 \sigma_a Q \tau_3 \sigma_b Q$$

$$+ \frac{D}{4}P_a T_{jk}\sigma_a \tau_3 \partial_j Q \partial_k Q + \frac{iD}{4}P_a K_{jk}\sigma_a \tau_3 Q \partial_j Q \partial_k Q \bigg). \tag{58}$$

By variation of the action with respect to $Q$, we find that the Usadel equation, expressed in terms of currents and torques, reads

$$\partial_k J_k = [g, \hat{\omega}_{t,t'}\tau_3 + \hat{\Delta}] + \Gamma_{ab}[g, \tau_3 \sigma_a g \tau_3 \sigma_b] + \mathcal{T}, \tag{59}$$

where the matrix current and torque of the system are given by

$$J_k = -Dg\partial_k g + \frac{D}{4}P_a T_{jk}\{\tau_3 \sigma_a + g\tau_3 \sigma_a g, g\partial_j g\} + \frac{iD}{4}P_a K_{jk}[\tau_3 \sigma_a + g\tau_3 \sigma_a g, \partial_j g], \tag{60}$$

while the torque becomes

$$\mathcal{T} = \frac{D}{4}P_a T_{jk}[\tau_3 \sigma_a, \partial_j g \partial_k g] + \frac{iD}{4}P_a K_{jk}[\tau_3 \sigma_a, g\partial_j g \partial_k g]. \tag{61}$$

Depending on the specific symmetry that ensures vanishing net magnetization, we may distinguish several different types of collinear magnetic structures without net exchange field. These are materials with crystal structures invariant under the product of the time-reversal ($\mathcal{T}$) with either translation, inversion, or a rotation [7]. In the former two cases, the material is called an antiferromagnet, in the latter case it is called an altermagnet.

Terms like those proportional to $T_{jk}$ and $K_{jk}$ in the action in Eq. (58), are first order in $\tau_3 \sigma_a$ and second order in real space derivatives, and hence they are even under inversion and odd under time-reversal. In other words, they can not appear in materials where an inversion compensates time-reversal. Moreover, since the NLSM describes fields on a mesoscopic scale, it is unaltered by translations on a microscopic scale. Thus, in antiferromagnets in which a translation compensates time-reversal, all terms that are odd in $\mathcal{T}$, i.e. that have an odd number of $\tau_3$'s vanish. Consequently, for any antiferromagnet, $T_{jk} = K_{jk} = 0$ in the action in Eq. (58). Thus, compared to a normal metal, the NLSM action of antiferromagnets contains only one additional term, related to spin-relaxation [$\Gamma_{ab}$ in Eq. (58)]. By using a specific microscopic model of antiferromagnetism, Ref. [27] showed that such a relaxation term may appear due to virtual transitions to bands far away from the Fermi level. In Ref. [101] it was shown that in altermagnets and, in fact, in any system with momentum dependent spin-splitting, the spin relaxation appears naturally, in analogy with the Dyankonov-Perel relaxation mechanism for spin-orbit coupling. To higher order in both $\tau_3 \sigma$ there exist additional terms in

the action that describe the difference in diffusion constant that may appear for spins parallel or perpendicular to the collinear axis of antiferromagnets [27,83].

Altermagnets are invariant under the combination of time-reversal with a rotation instead of a translation or an inversion. This difference has important consequences for the transport equations, because it allows the tensor $K_{jk}$ and $T_{jk}$ to be nonzero, and only imposes relations between the different components of those tensors. In general, there are 10 spin space groups which allow for collinear altermagnetism [5], see the second column in Table 1. We remind that the left superscript (1 or 2) of a point group element $\mathcal{R}$ acting on the spatial coordinates as $x \to \mathcal{R}x$, indicates whether the transformation of space goes together with spin-flip ($^2\mathcal{R}$), or acts on the spins trivially ($^1\mathcal{R}$). As the spin-dependent gradient terms in the action Eq. (58) contain the time reversal odd factor $\tau_3\boldsymbol{\sigma}$, the invariance under the spin group operation of type $^2\mathcal{R}$ implies antisymmetry of tensors $T$ and $K$ under its space part $\mathcal{R}$: $T = -\mathcal{R}T\mathcal{R}^{-1}$ and $K = -\mathcal{R}K\mathcal{R}^{-1}$. On the other hand, the symmetry with respect to the element of type $^1\mathcal{R}$, involving the identity operation in the spin space, translates to the conditions $T = \mathcal{R}T\mathcal{R}^{-1}$ and $K = \mathcal{R}K\mathcal{R}^{-1}$. This has to hold for all operations $^1\mathcal{R}$ and $^2\mathcal{R}$ in the spin space group.

In Appendix D we use the crystal symmetries to derive the allowed form of the tensors for each of these 10 spin space groups. The results are summarized in Table 1, where we used the real space basis characteristic of the real space lattice vectors. If another basis is used, the form of $T_{jk}$ and $K_{jk}$ has to be appropriately altered using the standard transformation rules for matrices in real space. For example, if a material with spin space group $^2m^2m^1m$ is used, but the chosen axes are rotated $\pi/4$ compared to the standard one, the only allowed nonzero components are $T_{xx} = -T_{yy}$. We find that the tensors $T_{jk}, K_{jk}$ can be nonzero for collinear magnetic structures without a net exchange field if and only if the material is a $d$ - wave altermagnet, for which the structure of $T_{jk}$ and $K_{jk}$ is derived in Appendix D.1. For $g$ - wave and $i$ - wave altermagnets, as discussed in Appendices D.2, D.3, the only effect on transport is the existence of spin relaxation, they are indistinguishable from antiferromagnets in their dirty limit transport properties. Therefore, from this point onwards we focus on $d$ - wave altermagnets.

## 5.1 Diffusion equation for $d$ - wave altermagnets

In this section we derive the transport equations in diffusive altermagnets from Eqs. (59-61). We first focus on the normal state. Following the same procedure as for ferromagnets, detailed in Sec. 4.1, we arrive at the diffusion equations for altermagnets:

$$\partial_t \delta n + \partial_k j_k = 0 \,, \tag{62}$$

$$\partial_t \delta S_b + \partial_k j_{ka}^s = -\Gamma_{ab} \delta S_b \,, \tag{63}$$

with

$$j_k = -\sigma_D \partial_k \mu - \sigma_D P_a T_{jk} \partial_j \mu_a^s \,, \tag{64}$$

$$j_{ka}^s = -\sigma_D \partial_k \mu_a^s - \sigma_D P_a T_{jk} \partial_j \mu - \sigma_D P_b K_{jk} \varepsilon_{abc} \partial_j \mu_c^s \,. \tag{65}$$

These equations are another important result of our work. They describe the electronic transport of diffusive altermagnets. The symmetry of the underlying magnetic structure is encoded in the tensors $T_{jk}$ and $K_{jk}$, see Table 1, which allows for identifying magnetic symmetries from transport measurements in mesoscopic-sized devices.

As an example, if one applies an electric current $j_k$ to an altermagnet, from Eq. (65) we directly read that in the normal state a spin-current is generated from a charge current via the term $T_{jk}$ according to:

$$j_{ai}^s = P_a T_{ki} j_k \,. \tag{66}$$

Table 1: The different types of altermagnetic spin space groups and the corresponding allowed form of $T_{jk}$ and $K_{jk}$, which obey the same symmetry constraints. The real space basis used is determined by the conventional choice of real space lattice vectors, the form of $T_{jk}$ and $K_{jk}$ has to be appropriately altered when using a different basis. Only for $d$ - wave altermagnets the tensors $T_{jk}$ and $K_{jk}$ can be nonvanishing. The last column contains examples of conducting materials that belong to the mentioned spin space groups [84–89]. All so-far predicted $i$ - wave altermagnets are semiconductors and thus not suitable for transport measurements.

| Type | Spin space group | $T_{jk}, K_{jk}$ | Examples |
|---|---|---|---|
| $d$-wave | $^2m^2m^1m$ | $\begin{bmatrix} 0 & T_{xy} & 0 \\ T_{xy} & 0 & 0 \\ 0 & 0 & 0 \end{bmatrix}$ | $La_2CuO_4$ |
| | $^24/^1m$ | $\begin{bmatrix} T_{xx} & T_{xy} & 0 \\ T_{xy} & -T_{xx} & 0 \\ 0 & 0 & 0 \end{bmatrix}$ | $KRu_4O_8$ |
| | $^24/^1m^2m^1m$ | $\begin{bmatrix} 0 & T_{xy} & 0 \\ T_{xy} & 0 & 0 \\ 0 & 0 & 0 \end{bmatrix}$ | $RuO_2$ |
| | $^22/^2m$ | $\begin{bmatrix} 0 & 0 & T_{xz} \\ 0 & 0 & T_{yz} \\ T_{xz} & T_{yz} & 0 \end{bmatrix}$ | $NaPr_2OsO_6$ |
| $g$ - wave | $^14/^1m^2m^2m$ | 0 | $CoF_3$ |
| | $^13^2m$ | | |
| | $^26/^2m$ | | |
| | $^26/^2m^2m^1m$ | | |
| $i$ - wave | $^16/^1m^2m^2m$ | 0 | - |
| | $^1m^13^2m$ | | |

This spin current is spin-polarized in the direction of the altermagnet collinear axis. However, its real space direction and magnitude depend on the direction of the charge current. The spin-current may have a longitudinal components, like in the ferromagnet, but also transverse components, if the direction of the charge current is not an eigenvector of $T_{jk}$. As shown in Appendix E, transversal spin currents lead to the normal state spin-splitter effect described for clean systems in Ref. [90], in which spins with opposite orientation move to opposite sides of the material, giving rise to spin accumulations on both sides, similar to the spin Hall effect. These spin accumulations are odd under current reversal.

We now focus on the term proportional to $K_{jk}$ in Eq. (65). In a ferromagnet, this term leads to a correction to the Hanle-like term proportional to the exchange field and was neglected in Section 4.1. However, in altermagnets the effective exchange field vanishes, and this term is

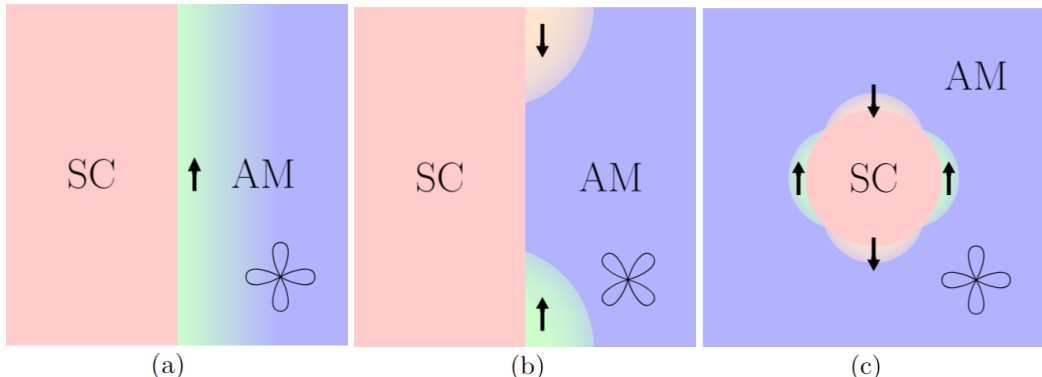

Figure 2: Three types of S / AM junctions in which a local magnetization is generated, indicated using green (spin up) and orange (spin down) colors. *(a):* The normal to the interface corresponds to a node direction of the altermagnet, which results in an edge magnetization. *(b):* the normal to the interface corresponds to a nodal direction of the altermagnet. There is no edge magnetization, only a corner magnetization *(c):* SC island on top of an altermagnet. A magnetic moment is induced along the interface between the superconductor and altermagnet, whose sign depends on the position along the perimeter.

the leading order of the spin Hanle effect. It describes the precession of spins polarized along noncollinear axes of the altermagnet.

To reveal such a precession, one can use a nonlocal spin valve as the ones used to detect the spin Hall effect [73] in normal metals. It involves electrically injecting spins from a ferromagnet into an altermagnet by passing a charge current, and then measuring the nonlocal voltage at a second ferromagnetic electrode used as a detector. The distance between the injector and detector needs to be smaller than the spin-relaxation length. The nonlocal signal depends on the distance between the injector and detector, the orientation of the wire axis to the altermagnet crystallographic axis, and the polarization direction of the injected spins. In contrast to normal metals, in an altermagnet, the spin precession occurs even in the absence of an applied magnetic field. Equations (62-65) provide a complete tool to describe experiments of this sort.

In short, Eqs. (62-65) represent new drift-diffusion equations for altermagnets that capture various magnetoelectric effects, which can be measured in experiments, enabling the study of underlying symmetries in magnetic materials through transport measurements. To the best of our knowledge, these equations are unprecedented.

## 5.2 Superconductivity and altermagnets

We now exploit the fact that the action in Eq. (58) and Usadel equation in Eqs. (59-61) are also valid in the superconducting case, and address altermagnets proximitized by superconductors or superconducting altermagnets.

We first consider an altermagnet with one single domain. We choose the collinear axis of the altermagnet to be along the z-direction. We assume a weak proximity effect from a superconductor, such that we can linearized the Usadel equation, Eq. (59), with respect to the condensate function $f$. This situation was also explored in Ref. [30]. However, below we show that some predictions of this work differ from our findings.

To this end, we consider a boundary with a normal along the x-direction between a superconducting altermagnet that carries a phase gradient $-q$ in the y-direction and vacuum. We suppose that the boundary is oriented in such a way that $K_{xy}$ is the only nonzero component.

Since no current can leave the material, we obtain the following boundary condition from Eq. (60):

$$0 = \partial_x f_s + iP_z K_{xy}\partial_y f_t\,, \tag{67}$$

$$0 = \partial_x f_t + iP_z K_{xy}\partial_y f_s\,. \tag{68}$$

If the altermagnetic term is small, the singlet component is approximately equal to its bulk value $f_s e^{-iqy}$, while according Eq. (68), near the boundary triplets are generated with

$$f_t \sim qP_z K_{xy}\,. \tag{69}$$

Because both $P_a$ and $q$ are time-reversal odd, this is a time-reversal even quantity. Therefore, the triplets are time-reversal even. Therefore, they can not be responsible for a time-reversal odd magnetization.

This consideration is confirmed by the nonlinear Usadel equation, Eqs. (59-61). Indeed, as shown in Appendix F.1, for an altermagnet in the superconducting state with a phase gradient we find

$$M_z(q) = M_z(-q)\,, \tag{70}$$

that is, the magnetization is even under current reversal, in agreement with [91]. In other words, there is no a superfluid analog of the normal state spin splitter effect. Also, for homogeneous phase gradients the current induced magnetization is even in the transverse coordinate, that is, spins do not localize on opposite edges.

For the particular orientation, in which $K_{xy}$ is the only nonzero component, an even stronger statement can be made, as shown in Appendix F.2. Indeed, the equations for $g$ are invariant under $g \rightarrow \sigma_y g^T \sigma_y$, which means that if $K_{xy}$ is the only nonzero component, the magnetization satisfies

$$M_z = -ig\mu_B\frac{\pi\nu}{4}\mathrm{tr}(\sigma_z g) = -ig\mu_B\frac{\pi\nu}{4}\mathrm{tr}(\sigma_z\sigma_y g^T\sigma_y) = ig\mu_B\frac{\pi\nu}{4}\mathrm{tr}(\sigma_z g) = -M_z\,, \tag{71}$$

which is only possible if

$$M_z = 0\,. \tag{72}$$

Our results are also confirmed by the consideration of a Josephson junction, superconductor-altermangent-superconductor, in Appendix F.3, where we show that $M_z(\varphi) = M_z(-\varphi)$, with $\varphi$ the phase difference between the superconducting electrodes. That is, any induced magnetization is even with respect to the Josphson current; i.e. there is no spin-splitter effect.

In addition to phase gradients, spatially inhomogeneous systems, such as hybrid structures, may also exhibit gradients in the magnitude of the pair amplitude. These gradients are not associated with currents, but with pair breaking, and, in contrast to phase gradients, they are time-reversal symmetric. Therefore, by replacing $iq$ in Eq. (68) by $\partial_k|f|$ and keeping a more general structure of $K_{jk}$ and boundary normal $n_j$, we find the following expression for triplets induced near the boundary of an altermagnet:

$$f_t \sim in_j K_{jk}P_z\partial_k|f|\,. \tag{73}$$

Because $P_z$ is time-reversal odd, but $\partial_k|f|$ is time-reversal even, this is a time-reversal odd quantity, and $f_t = \tilde{f}_t$. Therefore, gradients of the condensate magnitude in an altermagnet should give rise not only to triplet correlations but also to spin accumulation. Such variation occurs, for example, in a superconductor-altermagnet bilayer (see Fig. 2), where, via the proximity effect, the superconducting condensate penetrates into the altermagnet and decays over the thermal coherence length.

To examine this proximity induced magnetization, we consider the proximity effect in an S/AM junction with different orientations of the lobes of the $d$-wave with respect to the normal of the interface. We always choose our axis such that the normal is in the x-direction, and consequently need to modify the structure of $T, K$ compared to Table 1. In the longitudinal orientation, in which the lobe of the altermagnet is along the normal to the interface, $K_{xx} \neq 0$, $K_{xy} = 0$, and thus the equations, denoting the magnitude of the pair potential by $|\Delta|$, read

$$D(1 \pm iP_z K_{xx} \text{sign}(\omega_n)) \partial_{xx} f_\pm = 2|\omega| f_\pm \,, \tag{74}$$

$$D(1 \pm iP_z K_{xx} \text{sign}(\omega_n)) \partial_x f_\pm = -\gamma_B \frac{2|\Delta|}{|\omega|} \,, \tag{75}$$

where $\gamma_B$ is the Kuprianov Luckichev boundary parameter [92] which is proportional to the inverse of the S/AM interface resistance. Meanwhile $\tilde{f} = f$. In terms of singlets and triplets, introduced as $f_\pm = f_s \pm i f_t \text{sign}(\omega_n)$ these may be written as

$$D(\partial_{xx} f_s - P_z K_{xx} \partial_{xx} f_t) = 2|\omega| f_s \,, \tag{76}$$

$$D(\partial_{xx} f_t + P_z K_{xx} \partial_{xx} f_s) = 2|\omega| f_t \,, \tag{77}$$

$$D(\partial_x f_s + P_z K_{xx} \partial_x f_t) = -\gamma_B \frac{2|\Delta|}{|\omega|} \,, \tag{78}$$

$$D(\partial_x f_t - P_z K_{xx} \partial_x f_s) = 0 \,. \tag{79}$$

From the last equation we infer that, for $K_{xx} \neq 0$, triplets are induced close to the interface between a superconductor and an altermagnet. For full expressions of these triplet pair amplitudes, see Appendix G.1, Eq. (G.6). Since the equations satisfy $f = \tilde{f}$, the triplets induce a magnetization, which reads to first order in $K_{xx}$:

$$M_z(x) = -ig\mu_B \pi \nu \text{tr}(\tau_3 \sigma_z g(x)) = -2g\mu_B P_z K_{xx} \frac{\pi \nu}{4} \gamma_B^2 \frac{|\Delta|^2 D}{(\pi T)^2} \sum_n \frac{1}{(2n+1)^3} e^{-\sqrt{\frac{2\pi T(2n+1)}{D}} 2x} \,. \tag{80}$$

Exactly at the interface the magnetization takes the value

$$M_z(x=0) = -\frac{7\zeta(3)}{4} g\mu_B P_z K_{xx} \pi \nu \gamma_B^2 D \frac{|\Delta|^2}{(\pi T)^2} \,, \tag{81}$$

and then decays away from the boundary over a length scale $\sqrt{\frac{2D}{\pi T}}$ due to the suppression of both singlet and triplet pair amplitudes, as shown in Appendix G.1, Eq. (G.8). Thus, the presence of triplets causes a nonzero spin accumulation, as illustrated in Fig. 2(a).

In the transverse orientation shown in Fig. 2(b), only $K_{xy}$ is nonzero. If the system is finite in the transverse direction, a finite spin accumulation is created at the transverse boundaries of the altermagnet next to the superconductor, as shown in Fig. 2(b). For a calculation of the magnitudes of the induced magnetization, see Appendix G.2. This magnetization is also localized at the S / AM interface, decaying over a distance of the order $\sqrt{\frac{2D}{\pi T}}$. If the junction is infinite in the transverse direction, the Green's function does not depend on the $y$-coordinate, and consequently $K_{xy}$ drops out of Eq. (60). This means the singlets are decoupled from the triplets, leading to zero spin density.

We stress that the proximity induced magnetization is entirely different from the spin-splitter effect that appears in the normal state. Indeed, the effect arises in the absence of a current. Thus, even in the absence of an external source and even though neither the superconductor nor the altermagnet has a magnetization on its own, a hybrid system containing the two materials may show an equilibrium magnetization, whose details depend on the orientation of the lobes of the altermagnet. Such effects have been reported to appear in normal state

nonmagnetic oxides due to surface reconstruction [93]. The effect predicted here however, does not involve any reconstruction, but is a direct consequence of the proximity of the two types of materials. We may understand the appearance of this effect by comparing the orientations of the lobes in Fig. 2(a,b). In the orientation of (b), the lobes with spin up and down have the same orientation with respect to the boundary, which means that the lobes remain equivalent, and their contributions to the net magnetization cancel out, that is, there is no edge magnetization. On the other hand, for orientation (a), the orientation of the lobes of alter-magnetism with respect to the boundary is considerably different. Therefore, the equivalence between the lobes is broken by the proximity effect and a net edge magnetization remains. A similar consideration explains why in (b) a corner magnetization appears.

In fact, if a superconducting island is placed on top of an altermagnet, as shown in Fig. 2(c), our theory predicts a finite magnetization at the edges of the island, see Appendix G.3 for details. In particular, if the size of the island is large compared to the coherence length, the boundaries locally resemble an infinite plane, and the induced magnetization follows the directions of the lobes of the altermagnetic order, as shown in Fig. 2(c). The magnetization on opposite sides of the island is the same, but opposite to the magnetization along perpendicular directions, reflecting the $d$-wave symmetry of the altermagnet. The magnetization vanishes along the node directions of the altermagnet.

Since the effect appears when the magnitude of the pair potentials varies in space, a magnetization does not only arise near SN interfaces, but also in and near the weak link of Josephson junctions, or in materials in which superconductivity is locally suppressed due to strain or temperature gradients.

The effects described above appear in monodomain altermagnets where singlets and triplets with zero spin projections exist. In multidomains, with different collinear axes, there are additional effects, because the other two pair amplitudes, the equal spin ones, can be created. Alternatively, they can be created by using an altermagnet and a ferromagnet whose polarization is not along the collinear axis of the altermagnet. In the linearized limit we obtain the following Usadel equations for those equal spin pair amplitudes.

$$D(\delta_{jk} + T_{jk})\partial_j\partial_k f_{\uparrow\uparrow} = 2|\omega|f_{\uparrow\uparrow}, \tag{82}$$

$$D(\delta_{jk} - T_{jk})\partial_j\partial_k f_{\downarrow\downarrow} = 2|\omega|f_{\downarrow\downarrow}. \tag{83}$$

These equations are very similar to the equations for the spins in the normal state, and we find that for nonzero $T_{xy}$ the equal spin pairs, if created, localize on opposite sides and generate a spin accumulation. On the other hand, if only $T_{xx}$ is nonzero, only one type of spin may pass through the altermagnet. This means that in a Josephson junction with two altermagnets with different polarization axes (S/AM$_1$/AM$_2$/SC), the supercurrent is spin-polarized, in analogy with the SC/F$_1$/F$_2$/SC junctions discussed in Sec. 4. Moreover, since altermagnets break time-reversal symmetry, and using the three different domains inversion symmetry can be broken, anomalous currents may be generated in the same manner as for a ferromagnet. However, in an AM the spin dependence of the diffusion constant strongly depends on the direction in real space. Indeed, we know that the material is invariant under a $\pi/2$ rotation accompanied with a spin flip. Since a spin-flip changes the direction of the anomalous current, this means that if along the x-direction the anomalous current is away from one of the superconducting electrodes, in the y-direction it flows towards it. Therefore, at an angle $\pi/4$ the anomalous current vanishes. This emphasizes the importance of the crystal orientation of the altermagnet in such junctions.

# 6  *p* - wave magnets

In this section, we present our last example: the recently predicted *p*-wave magnets [6,94–96]. These materials are similar to antiferromagnets, as they are invariant under the combination of time-reversal and translation ($\mathcal{T}\boldsymbol{t}$). However, they are noncollinear and break inversion symmetry. In such materials, an odd-in-momentum splitting of the bands may arise, similar to what occurs in spin-orbit coupling [6].

Following our discussion of antiferromagnets in Sec. 5, if a translation compensates the time-reversal, all terms that are odd under $\mathcal{T}$, i.e., those that contain an odd number of $\tau_3$'s, vanish. Consequently, in the action given by Eq. (22), the terms proportional to $h_a$, $\chi_{ajk}$, and $\gamma_{ajk}$ all vanish. Thus, besides the relaxation term, the lowest-order invariants induced by the exchange correlations are those of the second order in $\tau_3\sigma_a$ and first order in derivatives. Apparently, the presence of the first order derivative requires the broken inversion symmetry, that is a characteristic feature of *p*-wave magnets. The charge conjugation and the chronology symmetry conditions allow two terms with the above anticipated structure, and thus we find that the effective action for *p*-wave magnets reads

$$
iS_P = \frac{\pi\nu}{2}\mathrm{Tr}\bigg( \frac{D}{4}(\nabla Q)^2 + \hat{\omega}_{t,t'}\tau_3 Q + \hat{\Delta}Q + \frac{1}{8}\Gamma_{ab}\tau_3\sigma_a Q\tau_3\sigma_b Q
$$
$$
+ \frac{1}{8}\lambda_{abj}\partial_j(\tau_3\sigma_a Q\tau_3\sigma_b Q) + \frac{1}{8}\varepsilon_{abc}\beta_{kc}\tau_3\sigma_a Q\tau_3\sigma_b Q\partial_k Q\bigg). \tag{84}
$$

The last two invariants entering the action with real tensor coefficients $\lambda_{abj}$ and $\beta_{kc}$ are expected to be responsible for the physics of *p*-wave magnets. As the term containing $\lambda_{abk}$ is a total derivative, its only effect is to modify the spin relaxation at boundaries. We therefore omit it in the following. Instead, we focus on the $\beta_{kc}$ term which brings qualitatively new physics. Following the symmetry arguments introduced in Sec. 5, the tensor $\beta_{kc}$ is nonzero only in noncollinear magnets. In coplanar magnets, it can be represented as $\beta_{ka} = P_a n_k$, where $P_a$ is a vector in spin space perpendicular to all spins in the material, and $n_k$ is a real space vector.

Since microscopically both *p*-wave magnets and inversion asymmetric materials with SOC, such as Rashba systems, are characterized by an odd-in-momentum spin splitting of the electronic bands, their properties are similar in many ways [6,16]. This similarity can also be seen from NLSM of Eq. (84). Indeed, the $\beta$ term in Eq. (84) differs from the spin-galvanic term in systems with spin-orbit coupling in [33] only because of the appearance of $\tau_3$'s next to the spin-Pauli matrices. Therefore, in the normal state, where $g$ commutes with $\tau_3$, the spin-galvanic effect in *p*-wave magnets is indistinguishable from the spin-galvanic effect in systems with SOC. In contrast, as we show below, this similarity breaks down in the superconducting state.

Let us now examine the consequences of the last term in the action Eq. (84) on the transport properties of a *p*-wave magnet. We analyze both superconducting and normal states and compare the results with those obtained in Ref. [33] for gyrotropic materials with SOC. By the variation of the action with respect to $Q$, we find the following Usadel equation

$$
\partial_k J_k = [g, \hat{\omega}_{t,t'}\tau_3 + \hat{\Delta}] + \Gamma_{ab}[g, \tau_3\sigma_a g\tau_3\sigma_b] + \mathcal{T}, \tag{85}
$$

where the matrix current is given by

$$
J_k = -Dg\partial_k g + \frac{i}{16}\varepsilon_{abc}\beta_{kc}\{[\tau_3\sigma_a, g], \tau_3\sigma_b + g\tau_3\sigma_b g\}, \tag{86}
$$

and the torque

$$
\mathcal{T} = -\frac{i}{8}\varepsilon_{abc}\beta_{kc}[\{\partial_k g, g\tau_3\sigma_a g\}, \sigma_b]. \tag{87}
$$

We consider a *p*-wave magnet with homogeneous superconducting potential. This pair potential can either be intrinsic or, if the *p*-wave magnet is much thinner than the superconducting coherence length, arise due to the proximity effect of a superconductor. Additionally, there is an externally induced homogeneous Zeeman field $\boldsymbol{h}$ perpendicular to the spins in the *p* - wave magnet. We choose the spins of the *p* - wave magnet to be in the xy-plane and $\boldsymbol{h}$ to be in the z-direction, with magnitude $h$. If the material is infinite in all direction, $g$ is independent of position and equals

$$
\begin{aligned}
g = {}& \frac{1}{2}(1+\sigma_z)\frac{1}{\sqrt{(\omega+ih)^2+|\Delta|^2}}((\omega+ih)\tau_3+|\Delta|\tau_2 e^{i\tau_3\varphi}) \\
& + \frac{1}{2}(1-\sigma_z)\frac{1}{\sqrt{(\omega-ih)^2+|\Delta|^2}}((\omega-ih)\tau_3+|\Delta|\tau_2 e^{i\tau_3\varphi}),
\end{aligned}
\tag{88}
$$

independent of the strength of *p* - wave magnetism. Evaluating the spin $\delta S_z$ following Eq. (39) and the current, combining Eqs. (40) and (86), to first order in the strength of the exchange field, we have

$$
\delta S_z = h2\pi T\sum_{\omega_n}\frac{|\Delta|^2}{(\omega_n^2+|\Delta|^2)^{\frac{3}{2}}},
\tag{89}
$$

$$
j_x = \beta_{xz}h2\pi T\sum_{\omega_n}\frac{|\Delta|^2\omega_n^2}{(\omega_n^2+|\Delta|^2)^{\frac{5}{2}}}.
\tag{90}
$$

Or, equivalently

$$
j_x = \beta(T)\delta S_z,
\tag{91}
$$

$$
\beta(T) = \beta_{xz}\frac{\sum_{\omega_n}\frac{\omega_n^2}{(\omega_n^2+|\Delta|^2)^{\frac{5}{2}}}}{\sum_{\omega_n}\frac{1}{(\omega_n^2+|\Delta|^2)^{\frac{3}{2}}}}.
\tag{92}
$$

Thus, anomalous currents are generated in the presence of a spin accumulation in a superconducting *p* - wave magnet, and the coefficient governing this effect depends on temperature. For lower temperatures the coupling between current and spin becomes weaker, although it does not vanish, but instead converges to $\frac{1}{3}$ of the normal state spin-galvanic coefficient as the temperature goes to zero. This temperature dependence of $\beta$ can be used to distinguish the spin-galvanic effect originating from *p*-wave magnetism from that induced by SOC, for which the spin-galvanic coefficient is independent of temperature. Thus, the difference arises at low temperatures where the equations linearized in pair amplitudes can not be used. This highlights the importance of using the nonlinear Usadel equation to describe low-temperature effects.

# 7 Conclusions

We have presented the quasiclassical transport theory for materials with magnetic exchange and superconducting correlations. Being derived from the Keldysh nonlinear sigma model, our theory describes transport at arbitrary temperatures, both in the normal and superconducting states, as well as in equilibrium and nonequilibrium situations. We focus on different types of magnetic ordering and discuss various effects that appear in ferromagnets, antiferromagnets, d-wave antiferromagnets, and p-wave magnets, both in the normal state and the superconducting state.

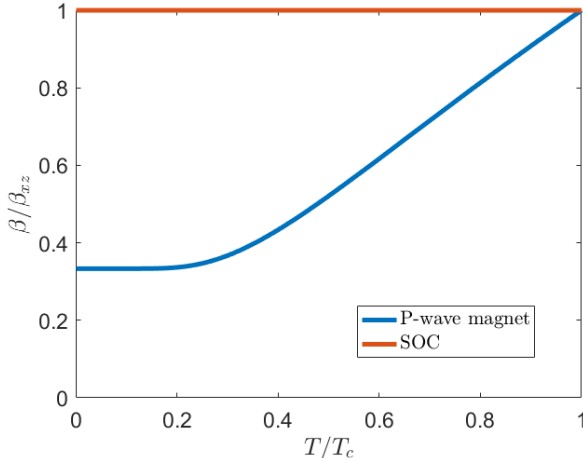

Figure 3: The magnitude of the spin-galvanic coefficient in the superconducting state normalized its the normal state value, induced by $p$ - wave magnetism or SOC. For SOC there is no difference between the normal state and superconducting state, $\beta$ does not depend on temperature, while the exchange induced spin-galvanic effect is weaker in the superconducting state than in the normal state, it varies continuously and converges to 1/3 of its normal state value at low temperatures.

For example, our diffusive equation for the quasiclassical Green's functions captures the long-sought spin-dependent diffusion coefficient within quasiclassical theory. Our results show that, in the normal state, a charge current is automatically accompanied by a spin current. On the other hand, in the superconducting state, this is not the case for supercurrents, unless equal-spin triplets are generated through a different mechanism.

We also show that the spin-splitting effect, predicted in clean systems [90], appears in the normal state of altermagnets, even if the material is diffusive. In the superconducting state, however, this effect vanishes, as dictated by the symmetry of our equations. The symmetry allows only for magnetization that is even in the phase gradient, which contrasts to the prediction in [30], but agrees with the clean limit results of Ref. [91]. Instead, our theory predicts another experimentally verifiable effect: the proximity induced magnetization, that is, the generation of a magnetization near interfaces between superconductors and altermagnets, due to the spatial dependence of the pair amplitude magnitude in the absence of supercurrents.

Next to this, we show that diffusive $p$ - wave magnets exhibit spin-galvanic effects and find that in the normal state, these effects are indistinguishable from spin-galvanic effects induced by SOC. In contrast, in the superconducting state they behave qualitatively different; we show that in $p$-wave magnets, in contrast to materials with SOC, the spin-galvanic coefficient is temperature dependent. In this way we provide a method to distinguish the two types of spin-galvanic effects.

Our symmetry-based derivation of the low-energy action sets the foundation for further developments in deriving kinetic equations for multiband systems, magnetic materials with strong magnetic interactions, and other hybrid systems. Specifically, it provides a framework for studying nonequilibrium effects in a wide variety of systems that combine superconductors, conventional magnets, and unconventional magnets. Moreover, our approach can be extended to multiband materials with additional electronic degrees of freedom.

# Acknowledgments

We thank A.A. Golubov, A. Mazanik and R. de las Heras for useful discussions.

**Funding information** We acknowledge financial support from Spanish MCIN/AEI/ 10.13039/501100011033 through projects PID2023-148225NB-C31, PID2023-148225NB-C32(SUNRISE), and TED2021-130292B-C42, the Basque Government through grant IT-1591-22, and European Union's Horizon Europe research and innovation program under grant agreement No 101130224 (JOSEPHINE) and Grupos Consolidados UPV/EHU del Gobierno Vasco (Grant IT1453-22).

**Note** During submission of our manuscript other articles addressing transport properties of altermagnets appeared in arXiv, discussing the derivation of the normal state diffusion equations and spin-transfer from a microscopic tight-binding model [97], and a phenomenological approach to the spin-transfer torque in normal state altermagnets [98]. Moreover, a few articles discussing the interplay of superconductivity and p-wave magnetism in the clean limit appeared [99, 100].

# A    Construction of the effective action

In this Appendix, we show how to construct the general action, Eq. (22) for collinear magnets, and Eq. (84) for p-wave magnets. Specifically, we construct the terms in the action by considering the lowest orders in derivatives in $\tau_3\boldsymbol{\sigma}$. The action is a scalar, and therefore these resulting terms need to be contracted with tensors, whose structures depend on the spin and spatial symmetries of the material under consideration. As explained in the main text, the number of allowed terms is reduced by recognizing that several terms can be transformed into one another and are hence equivalent, using the following operations: (i) cyclic permutation of the trace, (ii) the soft-mode normalization condition $Q^2 = \mathbf{1}$ and (iii) the identity $Q\partial_j Q = -\partial_j Q Q$, which is a consequence of (ii) holding throughout space.

Once all the different terms are identified, we apply the charge conjugation symmetry to determine which components of the tensor are allowed to be nonzero, and the chronology symmetry to identify whether the coefficients are real or imaginary.

In the next subsections, we explore the different terms up to second order in the exchange field and various orders in derivatives.

## A.1    First order in exchange

First we consider those terms up to first order in $\tau_3\boldsymbol{\sigma}$ and thus the tensor that is required is a vector in spin space. They are time-reversal odd and change sign upon reversal of all spins.

### A.1.1    Zeroth order in derivatives: Usual exchange term in ferromagnets and Zeeman field

The first order term in $\tau_3\boldsymbol{\sigma}$ without any derivative is:

$$\text{Tr}\left(ih_a \sigma_a \tau_3 Q\right). \tag{A.1}$$

This is the only term we may write to this order, because any additionally added $Q$'s can be eliminated via $Q^2 = \mathbf{1}$. Applying charge conjugation symmetry, i.e. $Q = \rho_1 \tau_1 \sigma_y Q^T \sigma_y \tau_1 \rho_1$

we find the constraint

$$\text{Tr}(ih_a\sigma_a\tau_3 Q) = \text{Tr}(h_a\sigma_a\tau_3\rho_1\tau_1\sigma_y Q^T\sigma_y\tau_1\rho_1) = -i\text{Tr}(h_a\sigma_y\sigma_a\sigma_y\tau_3 Q^T)$$
$$= i\text{Tr}(h_a\sigma_a^T\tau_3 Q^T) = i\text{Tr}(h_a Q\sigma_a\tau_3) = i\text{Tr}(h_a\sigma_a\tau_3 Q), \tag{A.2}$$

where we used $\tau_1\tau_3\tau_1 = -\tau_3$, $\sigma_y\sigma_a\sigma_y = \sigma_a^T$ and invariance of the trace under both transposition and cyclic permutation. Thus, charge conjugation symmetry is satisfied without any constraint on $h_a$.

Chronology symmetry, $iS[Q] = (iS[-\rho_2\tau_3 Q^\dagger\tau_3\rho_2])^*$, implies that

$$\text{Tr}\Big(ih_a\sigma_a\tau_3 Q\Big) = \left(i\text{Tr}\Big(h_a\sigma_a\tau_3(-\rho_2\tau_3 Q^\dagger\tau_3\rho_2)\Big)\right)^* = \left(-i\text{Tr}\Big(h_a\sigma_a\tau_3 Q^\dagger\Big)\right)^*$$
$$= i\text{Tr}\Big((h_a)^* Q\sigma_a\tau_3\Big) = i\text{Tr}\Big((h_a)^*\tau_3\sigma_a Q\Big), \tag{A.3}$$

where we used that the trace of the Hermitian conjugate of an expression is the complex conjugate of the trace of this expression. Thus, chronology symmetry implies $h_a$ is real.

### A.1.2 First order in derivatives

We may write down two terms with a single derivative, which contain second rank tensors with one spin index and one real space index:

$$i\text{Tr}\Big(\alpha_{aj}\sigma_a\tau_3\partial_j Q\Big), \tag{A.4}$$

$$\text{Tr}\Big(\kappa_{aj}\sigma_a\tau_3 Q\partial_j Q\Big). \tag{A.5}$$

These are the only two possible terms, since any additional $Q$'s can be eliminated via cyclic permutation of the trace, and the identities $Q\partial_j Q = -\partial_j QQ$ and $Q^2 = \mathbf{1}$. The terms, Eqs.(A.4-A.5) break both inversion and time-reversal, but are invariant under the product of these two operations. By charge conjugation the first term has to satisfy

$$i\text{Tr}\Big(\alpha_{aj}\sigma_a\tau_3\partial_j Q\Big) = i\text{Tr}\Big(\alpha_{aj}\sigma_a\tau_3\rho_1\tau_1\sigma_y\partial_j Q^T\sigma_y\tau_1\rho_1\Big)$$
$$= -i\text{Tr}\Big(\alpha_{aj}\sigma_y\sigma_a\tau_3\sigma_y\partial_j Q^T\Big) = i\text{Tr}\Big(\alpha_{aj}\sigma_a^T\tau_3\partial_j Q^T\Big)$$
$$= i\text{Tr}\Big(\alpha_{aj}\partial_j Q\sigma_a\tau_3\Big) = i\text{Tr}\Big(\alpha_{aj}\sigma_a\tau_3\partial_j Q\Big). \tag{A.6}$$

This is automatically satisfied, and therefore this term is allowed. Chronology symmetry for this term implies

$$\text{Tr}\Big(i\alpha_{aj}\sigma_a\tau_3\partial_j Q\Big) = \left(i\text{Tr}\Big(\alpha_{aj}\sigma_a\tau_3(-\rho_2\tau_3\partial_j Q^\dagger\tau_3\rho_2)\Big)\right)^*$$
$$= \left(-i\text{Tr}\Big(\alpha_{aj}\sigma_a\tau_3\partial_j Q^\dagger\Big)\right)^* = i\text{Tr}\Big(\alpha_{aj}^*\partial_j Q\sigma_a\tau_3\Big)$$
$$= i\text{Tr}\Big(\alpha_{aj}^*\tau_3\sigma_a\partial_j Q\Big), \tag{A.7}$$

which means that $\alpha_{aj}$ is a real second rank tensor with one spin and one real space index. However, as explained in the main text, this term is a total derivative, and therefore we disregarded from the action, Eq. (22).

The second term, Eq. (A.5), does not enter the action. Indeed, as explained in the main text, charge conjugation symmetry implies $\kappa_{aj} = 0$, see Eq. (18). in Sec. 2.

### A.1.3 Second order in derivatives: Novel terms in the action of collinear ferromagnets and altermagnets

Next, we consider terms linear in $\tau_3\boldsymbol{\sigma}$ with two derivatives. These terms contain third rank tensors with one spin index and two real space indices. There exist two such terms, corresponding to Eqs. (19) and (20) in the main text.

$$\mathrm{Tr}\!\left(\gamma_{ajk}\sigma_a\tau_3\partial_jQ\partial_kQ\right), \tag{A.8}$$

$$i\mathrm{Tr}\!\left(\chi_{ajk}\sigma_a\tau_3Q\partial_jQ\partial_kQ\right). \tag{A.9}$$

Again, adding more $Q$'s does not yield any new terms, as can be shown using cyclic permutation of the trace and the identities $Q\partial_jQ = -\partial_jQQ$ and $Q^2 = \mathbf{1}$.

Charge conjugation on the first term leads to the constraint

$$
\begin{aligned}
\mathrm{Tr}\!\left(\gamma_{ajk}\sigma_a\tau_3\partial_jQ\partial_kQ\right) &= \mathrm{Tr}\!\left(\gamma_{ajk}\sigma_a\tau_3\rho_1\tau_1\sigma_y\partial_jQ^T\sigma_y\tau_1\rho_1\rho_1\tau_1\sigma_y\partial_kQ^T\sigma_y\tau_1\rho_1\right) \\
&= -\mathrm{Tr}\!\left(\gamma_{ajk}\sigma_y\sigma_a\sigma_y\tau_3\partial_jQ^T\partial_kQ^T\right) = \mathrm{Tr}\!\left(\gamma_{ajk}\sigma_a^T\tau_3\partial_jQ^T\partial_kQ^T\right) \\
&= \mathrm{Tr}\!\left(\gamma_{ajk}\partial_kQ\partial_jQ\sigma_a\tau_3\right) = \mathrm{Tr}\!\left(\gamma_{ajk}\sigma_a\tau_3\partial_kQ\partial_jQ\right) \\
&= \mathrm{Tr}\!\left(\gamma_{akj}\sigma_a\tau_3\partial_jQ\partial_kQ\right),
\end{aligned} \tag{A.10}
$$

while charge conjugation on the second term leads to the constraint

$$
\begin{aligned}
i\mathrm{Tr}\!\left(\chi_{ajk}\sigma_a\tau_3Q\partial_jQ\partial_kQ\right) &= i\mathrm{Tr}\Big(\chi_{ajk}\sigma_a\tau_3\rho_1\tau_1\sigma_yQ^T\tau_1\rho_1\sigma_y\rho_1\tau_1\sigma_y\partial_jQ^T\sigma_y\tau_1\rho_1 \\
&\qquad\qquad \times \rho_1\tau_1\sigma_y\partial_kQ^T\sigma_y\tau_1\rho_1\Big) \\
&= -i\mathrm{Tr}\!\left(\chi_{ajk}\sigma_y\sigma_a\sigma_y\tau_3Q^T\partial_jQ^T\partial_kQ^T\right) = i\mathrm{Tr}\!\left(\chi_{ajk}\sigma_a^T\tau_3Q^T\partial_jQ^T\partial_kQ^T\right) \\
&= i\mathrm{Tr}\!\left(\chi_{ajk}\partial_kQ\partial_jQQ\sigma_a\tau_3\right) = i\mathrm{Tr}\!\left(\chi_{ajk}\sigma_a\tau_3Q\partial_kQ\partial_jQ\right) \\
&= i\mathrm{Tr}\!\left(\chi_{akj}\sigma_a\tau_3Q\partial_jQ\partial_kQ\right),
\end{aligned} \tag{A.11}
$$

where it was used that $Q$ anticommutes with both $\partial_jQ$ and $\partial_kQ$. Thus, for both both cases, charge conjugation interchanges the spatial indices without introducing a minus sign, and consequently both tensors are symmetric in the spatial indices, $\gamma_{ajk} = \gamma_{akj}$ and $\chi_{ajk} = \chi_{akj}$.

Applying chronology symmetry to the first term we find, using that the trace of the Hermitian conjugate of a matrix is the complex conjugate of the trace of the matrix itself, that

$$
\begin{aligned}
\mathrm{Tr}\!\left(\gamma_{ajk}\sigma_a\tau_3\partial_jQ\partial_kQ\right) &= \left(\mathrm{Tr}\!\left(\gamma_{ajk}\sigma_a\tau_3(-\rho_2\tau_3\partial_jQ^\dagger\tau_3\rho_2)(-\rho_2\tau_3\partial_kQ^\dagger\tau_3\rho_2)\right)\right)^* \\
&= \mathrm{Tr}\!\left(\gamma_{ajk}^*\partial_kQ\partial_jQ\tau_3\sigma_a\right) = \mathrm{Tr}\!\left(\gamma_{ajk}^*\sigma_a\tau_3\partial_kQ\partial_jQ\right) \\
&= \mathrm{Tr}\!\left(\gamma_{akj}^*\sigma_a\tau_3\partial_jQ\partial_kQ\right),
\end{aligned} \tag{A.12}
$$

which implies $\gamma_{ajk} = \gamma_{akj}^*$. Since this tensor is symmetric in its spatial indices this means it is real. Applying chronology symmetry to the second term we find

$$
\begin{aligned}
i\mathrm{Tr}\!\left(\chi_{ajk}\sigma_a\tau_3Q\partial_jQ\partial_kQ\right) &= \left(i\mathrm{Tr}\!\left(\chi_{ajk}\sigma_a\tau_3(-\rho_2\tau_3Q^\dagger\tau_3\rho_2)(-\rho_2\tau_3\partial_jQ^\dagger\tau_3\rho_2)(-\rho_2\tau_3\partial_kQ^\dagger\tau_3\rho_2)\right)\right)^* \\
&= \left(-i\mathrm{Tr}\!\left(\chi_{ajk}\sigma_a\tau_3Q^\dagger\partial_jQ^\dagger\partial_kQ^\dagger\right)\right)^* = i\mathrm{Tr}\!\left(\chi_{ajk}^*\partial_kQ\partial_jQQ\tau_3\sigma_a\right) \\
&= i\mathrm{Tr}\!\left(\chi_{ajk}^*\sigma_a\tau_3\partial_kQ\partial_jQQ\right) = i\mathrm{Tr}\!\left(\chi_{akj}^*\sigma_a\tau_3Q\partial_jQ\partial_kQ\right),
\end{aligned} \tag{A.13}
$$

that is $\chi_{ajk} = \chi_{akj}^*$. Since $\chi_{ajk}$ is symmetric in its spatial indices $j,k$, it is real.

## A.2 Second order in exchange

Next, we consider those terms that are second order in $\tau_3\boldsymbol{\sigma}$. These terms, even though they can only be present if there are time-reversal symmetry breaking mechanisms in the materials, are themselves even in time-reversal, they do not change upon reversal of all spins.

### A.2.1 Zeroth order in derivatives: Spin relaxation term

Without derivatives, we may write down

$$\text{Tr}\Big(\Gamma_{ab}\tau_3\sigma_a Q\tau_3\sigma_b Q\Big),\tag{A.14}$$

By cyclic permutation of the trace the term is symmetric in the spin indices, i.e. $\Gamma_{ab} = \Gamma_{ba}$. which is the usual spin-relaxation term in magnetic structures. By charge conjugation we find the constraint

$$
\begin{aligned}
\text{Tr}\Big(\Gamma_{ab}\tau_3\sigma_a Q\tau_3\sigma_b Q\Big) &= \text{Tr}\Big(\Gamma_{ab}\tau_3\sigma_a\rho_1\tau_1\sigma_y Q^T\sigma_y\tau_1\rho_1\tau_3\sigma_b\rho_1\tau_1\sigma_y Q^T\sigma_y\tau_1\rho_1\Big)\\
&= \text{Tr}\Big(\Gamma_{ab}\tau_3\sigma_y\sigma_a\sigma_y Q^T\tau_3\sigma_y\sigma_b\sigma_y Q^T\Big)\\
&= \text{Tr}\Big(\Gamma_{ab}\tau_3\sigma_a^T Q^T\tau_3\sigma_b^T Q^T\Big) = \text{Tr}\Big(\Gamma_{ab}Q\tau_3\sigma_b Q\tau_3\sigma_a\Big)\\
&= \text{Tr}\Big(\Gamma_{ab}\tau_3\sigma_a Q\tau_3\sigma_b Q\Big),
\end{aligned}\tag{A.15}
$$

which means charge conjugation does not impose any additional restrictions on the tensor. No other terms with two $\tau_3\boldsymbol{\sigma}$'s and no derivatives can be constructed using cyclic permutation of the trace, and the normalization condition $Q^2 = \mathbb{1}$.

Chronology implies

$$
\begin{aligned}
\text{Tr}\Big(\Gamma_{ab}\tau_3\sigma_a Q\tau_3\sigma_b Q\Big) &= \Big(\text{Tr}\Big(\Gamma_{ab}\tau_3\sigma_a(-\rho_2\tau_3 Q^\dagger\tau_3\rho_2)\tau_3\sigma_b(-\rho_2\tau_3 Q^\dagger\tau_3\rho_2)\Big)\Big)^*\\
&= \text{Tr}\Big(\Gamma_{ab}\tau_3\sigma_a Q^\dagger\tau_3\sigma_b Q^\dagger\Big)^* = \text{Tr}\Big(\Gamma_{ab}^* Q\tau_3\sigma_b Q\tau_3\sigma_a\Big)\\
&= \text{Tr}\Big(\Gamma_{ab}^*\tau_3\sigma_a Q\tau_3\sigma_b Q\Big),
\end{aligned}\tag{A.16}
$$

that is, $\Gamma_{ab}$ is real.

### A.2.2 First order in derivatives: p-wave magnets

To first order in derivative, we require tensors with two spin-indices and one real space index. Consequently, we may write

$$\text{Tr}\Big(\lambda_{abj}\tau_3\sigma_a Q\tau_3\sigma_b\partial_j Q\Big),\tag{A.17}$$

$$\text{Tr}\Big(\kappa_{abj}\tau_3\sigma_a Q\tau_3\sigma_b Q\partial_j Q\Big).\tag{A.18}$$

No other terms with two $\tau_3$'s and two spin-Pauli matrices can be constructed using cyclic permutation of the trace, $Q\partial_j Q = -\partial_j QQ$ and $Q^2 = \mathbb{1}$. The charge conjugation constraints read

$$
\begin{aligned}
\text{Tr}\Big(\lambda_{abj}\tau_3\sigma_a Q\tau_3\sigma_b\partial_j Q\Big) &= \text{Tr}\Big(\lambda_{abj}\tau_3\sigma_a\rho_1\tau_1\sigma_y Q^T\sigma_y\tau_1\rho_1\tau_3\sigma_b\rho_1\tau_1\sigma_y\partial_j Q^T\sigma_y\tau_1\rho_1\Big)\\
&= \text{Tr}\Big(\lambda_{abj}\tau_3\sigma_y\sigma_a\sigma_y Q^T\tau_3\sigma_y\sigma_b\sigma_y\partial_j Q^T\Big)\\
&= \text{Tr}\Big(\lambda_{abj}\tau_3\sigma_a^T Q^T\tau_3\sigma_b^T\partial_j Q^T\Big) = \text{Tr}\Big(\lambda_{abj}\partial_j Q\tau_3\sigma_b Q\tau_3\sigma_a\Big)\\
&= \text{Tr}\Big(\lambda_{abj}\tau_3\sigma_b Q\tau_3\sigma_a\partial_j Q\Big),
\end{aligned}\tag{A.19}
$$

and

$$\begin{aligned}
\mathrm{Tr}\Big(\kappa_{abj}\tau_3\sigma_a Q\tau_3\sigma_b Q\partial_j Q\Big) &= \mathrm{Tr}\Big(\kappa_{abj}\tau_3\sigma_a\rho_1\tau_1\sigma_y Q^T\sigma_y\tau_1\rho_1\tau_3\sigma_b\rho_1\tau_1\sigma_y Q^T\sigma_y\tau_1\rho_1 \\
&\qquad\qquad \times \rho_1\tau_1\sigma_y\partial_j Q^T\sigma_y\tau_1\rho_1\Big) \\
&= \mathrm{Tr}\Big(\kappa_{abj}\tau_3\sigma_a\sigma_y\sigma_a\sigma_y Q^T\tau_3\sigma_y\sigma_b\sigma_y Q^T\partial_j Q^T\Big) \\
&= \mathrm{Tr}\Big(\kappa_{abj}\tau_3\sigma_a^T Q^T\tau_3\sigma_b^T Q^T\partial_j Q^T\Big) = \mathrm{Tr}\Big(\kappa_{abj}\partial_j Q Q\tau_3\sigma_b Q\tau_3\sigma_a\Big) \\
&= -\mathrm{Tr}\Big(\kappa_{abj}\tau_3\sigma_b Q\tau_3\sigma_a Q\partial_j Q\Big).
\end{aligned} \tag{A.20}$$

Thus, charge conjugation implies that $\lambda_{abj}$ is even in its spin indices, while $\kappa_{abj}$ is odd in its spin indices. For the first term this implies we may write it as

$$\frac{1}{2}\lambda_{ajk}\partial_j(\tau_3\sigma_a Q\tau_3\sigma_b Q), \tag{A.21}$$

while for the latter it means we may write

$$\mathrm{Tr}\Big(\varepsilon_{abc}\beta_{kc}\tau_3\sigma_a Q\tau_3\sigma_b Q\partial_j Q\Big). \tag{A.22}$$

These are the terms appearing in the action for the $p$ - wave magnet in Eq. (84).

Chronology symmetry implies

$$\begin{aligned}
\mathrm{Tr}\Big(\lambda_{abj}\tau_3\sigma_a Q\tau_3\sigma_b\partial_j Q\Big) &= \left(\mathrm{Tr}\Big(\lambda_{abj}\tau_3\sigma_a(-\tau_3\rho_2 Q^\dagger\rho_2\tau_3)\tau_3\sigma_b(-\tau_3\rho_2\partial_j Q\rho_2\tau_3)\Big)\right)^* \\
&= \left(\mathrm{Tr}\Big(\lambda_{abj}\tau_3\sigma_a Q^\dagger\tau_3\sigma_b\partial_j Q^\dagger\Big)\right)^* \\
&= \mathrm{Tr}\Big(\lambda_{abj}^*\partial_j Q\sigma_b\tau_3 Q\sigma_a\tau_3\Big) = \mathrm{Tr}\Big(\lambda_{abj}^*\tau_3\sigma_b Q\tau_3\sigma_a\partial_j Q\Big) \\
&= \mathrm{Tr}\Big(\lambda_{baj}^*\tau_3\sigma_a Q\tau_3\sigma_b\partial_j Q\Big),
\end{aligned} \tag{A.23}$$

that is, $\lambda_{abj} = \lambda_{baj}^*$. Since the tensor is even in the spin indices $a, b$, this implies that $\lambda_{abj}$ is real.

For the second order term we find

$$\begin{aligned}
i\mathrm{Tr}\Big(\beta_{jc}\varepsilon_{abc}\tau_3\sigma_a Q\tau_3\sigma_b Q\partial_j Q\Big) &= \Big(i\mathrm{Tr}\Big(\beta_{jc}\varepsilon_{abc}\tau_3\sigma_a(-\tau_3\rho_2 Q^\dagger\rho_2\tau_3) \\
&\qquad\qquad \times \tau_3\sigma_b(-\tau_3\rho_2 Q^\dagger\rho_2\tau_3)(-\tau_3\rho_2\partial_j Q\rho_2\tau_3)\Big)\Big)^* \\
&= -\Big(i\mathrm{Tr}\Big(\beta_{jc}\varepsilon_{abc}\tau_3\sigma_a Q^\dagger\tau_3\sigma_b Q^\dagger\partial_j Q^\dagger\Big)\Big)^* \\
&= i\mathrm{Tr}\Big((\beta)_{jc}^*\varepsilon_{abc}\partial_j Q Q\sigma_b\tau_3 Q\sigma_a\tau_3\Big) \\
&= -i\mathrm{Tr}\Big((\beta)_{jc}^*\varepsilon_{abc}\tau_3\sigma_b Q\tau_3\sigma_a Q\partial_j Q\Big) \\
&= i\mathrm{Tr}\Big((\beta)_{jc}^*\varepsilon_{abc}\tau_3\sigma_a Q\tau_3\sigma_b\partial_j Q\Big),
\end{aligned} \tag{A.24}$$

and therefore also $\beta_{jc}$ is real.

In summary, all previous subsections reproduce all terms in the actions Eq. (22, 84). We have shown that to lowest orders, there exist no other terms that are allowed in the action.

# B  The Usadel equation: Identification of the currents and torques from the action

In this section we discuss how to calculate the current and torque that appear in the Usadel equation from the action of the nonlinear sigma model, that is, how to derive Eqs. (23-25) from Eq. (22) in Sec. 3 of the main text, and Eqs. (31-33) from Eq. (30) in Sec. 4, and Eqs. (59-61) from Eq. (58) in Sec. 5, and Eqs. (85-87) from Eq. (84) in Sec. 6.

The Usadel equation is the saddle point equation of the NLSM. Therefore, it can be found by variation of the action that leave the normalization $Q^2 = 1$ intact. In this case perturbation of $Q$ can be expressed as $\delta Q = [\alpha, Q]$. Variation of the action leads to

$$\delta S = \int_\Omega dV \operatorname{Tr}\left( \frac{\delta S}{\delta Q} \delta Q + \frac{\delta S}{\delta \partial_j Q} \delta \partial_j Q \right), \tag{B.1}$$

where $\int_\Omega dV$ denotes integration over the entire volume. The stationary configuration of $Q$ is obtained by imposing $\delta S = 0$. After integration by parts we find

$$\int_\Omega dV \operatorname{Tr}\left( \frac{\delta S}{\delta Q} - \partial_j \frac{\delta S}{\delta \partial_j Q} \right) \delta Q + \int_{\partial \Omega} dS\, n_j \operatorname{Tr}\left( \partial_j \frac{\delta S}{\delta \partial_j Q} \right) \delta Q = 0, \tag{B.2}$$

where $\int_{\partial \Omega} dS$ denotes integration over the system boundary and $n_j$ denotes the local normal vector to the boundary. Using that $\delta Q = [\alpha, Q]$ we find

$$\int_\Omega dV \operatorname{Tr}\left( \frac{\delta S}{\delta Q} - \partial_j \frac{\delta S}{\delta \partial_j Q} \right)[\alpha, Q] + \int_{\partial \Omega} dS \operatorname{Tr}\left( \partial_j \frac{\delta S}{\delta \partial_j Q} \right)[\alpha, Q]$$
$$= \int_\Omega dV \operatorname{Tr}\left( \left[ Q, \frac{\delta S}{\delta Q} - \partial_j \frac{\delta S}{\delta \partial_j Q} \right] \alpha \right) + \int_{\partial \Omega} dS\, n_j \operatorname{Tr}\left( \left[ Q, \partial_j \frac{\delta S}{\delta \partial_j Q} \right] \alpha \right) = 0. \tag{B.3}$$

Since this must hold for any possible $\alpha$, the expressions multiplied by $\alpha$ must vanish themselves. In the bulk we obtain

$$\left[ Q, \frac{\delta S}{\delta Q} - \partial_j \frac{\delta S}{\delta \partial_j Q} \right] = -\partial_j \left[ Q, \frac{\delta S}{\delta \partial_j Q} \right] + \left[ Q, \frac{\delta S}{\delta Q} \right] + \left[ \partial_j Q, \frac{\delta S}{\delta \partial_j Q} \right] = 0.$$

Here we identify the matrix current defined as $J_j = i[Q, \frac{\delta S}{\delta \partial_j Q}]$ and the matrix torque $\mathcal{T} = i[Q, \frac{\delta S}{\delta Q}] + i[\partial_j Q, \frac{\delta S}{\delta \partial_j Q}]$. Thus, the bulk equation reads

$$\partial_j J_j = \mathcal{T}, \tag{B.4}$$

which is the continuity equation for the matrix current.

From second term in Eq. (B.3) we find at the boundary:

$$n_j [Q, \frac{\delta S}{\delta \partial_j Q}]\bigg|_{\partial \Omega} = 0, \tag{B.5}$$

which defines the boundary condition as

$$n_j J_j \big|_{\partial \Omega} = 0. \tag{B.6}$$

This condition reflects that no current can leave or enter the system if we do not introduce source or sink terms in the boundary action.

Both the current and the torque contain $Q$ evaluated at the saddle point, and are therefore often presented in terms of its saddle-point value, the quasiclassical Green's function $g$.

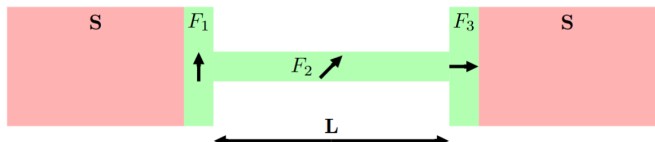

Figure 4: The S / $F_1$ / $F_2$ / $F_3$ / S junction studied in this Appendix, where the three ferromagnets have mutually orthogonal orientations. The middle ferromagnet is much thinner and longer than the other ferromagnets, so that these together with the superconductors can be treated as reservoirs.

### B.1 Matrix current from the vector potential

Above, we have identified $J_j$ as a matrix current. Here we show that $J_j$ really describes the physical currents. To this end, we use that the physical currents, i.e. the charge and spin currents of the system, are obtained from $-\frac{\delta S}{\delta \mathcal{A}}$, where $\mathcal{A}$ is the matrix vector potential that contains both the usual scalar electromagnetic potential and the vector electromagnetic potential. This matrix vector potential enters the covariant derivatives $\hat{\partial}_j \cdot = \partial_j \cdot -i[\mathcal{A}_j, \cdot]$ [32]. Thus, the observable matrix current $\mathcal{J}_j$ can be calculated from the action, keeping only terms linear in $\delta Q$:

$$\mathrm{Tr}\left(\mathcal{J}_j \mathcal{A}_j\right) := -\frac{\delta S}{\delta \mathcal{A}_j}\mathcal{A}_j = -\left(S(Q, \partial_j Q - i[\mathcal{A}_j, Q]) - S(Q, \partial_j Q)\right) = -\mathrm{Tr}\left(-i[\mathcal{A}_j, Q]\frac{\delta S}{\delta \partial_j Q}\right)$$

$$= i\mathrm{Tr}\left(\left[Q, \frac{\delta S}{\delta \partial_j Q}\right]\mathcal{A}_j\right) = \mathrm{Tr}\left(J_j \mathcal{A}_j\right). \tag{B.7}$$

In the last equality we use the definition of $J_j$ introduced in the previous section. From the fact that this must hold for any $\mathcal{A}_j$ we deduce that

$$J_j = \mathcal{J}_j, \tag{B.8}$$

and consequently $J_j$ is the observable matrix current of the system.

## C Anomalous currents in ferromagnetic structures

In this section we prove our claim in Sec. 4 of the main text. Namely, that from our Usadel equation, Eq. (31), one can describe anomalous currents in Josephson junctions in which the weak link consists of three magnetic domains with noncoplanar magnetizations, Eq. (57) in the main text.

Consider a ferromagnet with magnetization $\boldsymbol{m}_3$, which we choose to be the z-direction, that is proximitized sandwiched by two ferromagnets much thinner than the coherence length at $x = \pm\frac{L}{2}$, with magnetizations $\boldsymbol{m}_1$ and $\boldsymbol{m}_2$, and two superconducting leads at $|x| > \frac{L}{2}$.

For simplicity of calculation and notation, and to illustrate the effect, we assume that all contacts are good, so that $G$ is continuous, that the strength of the exchange field is the same in all ferromagnets and that the middle ferromagnet has a much smaller width than the outer ferromagnets, so that the latter can, together with the superconductors, be treated as reservoirs. This setup is illustrated in Fig. 4.

In this case, we may consider the outer ferromagnet/superconductor bilayers, as reservoirs with a BCS like density of states split by the corresponding exchange fields pointing in the direction of $\boldsymbol{m}_{1,2}$ respectively. At $x = \pm\frac{L}{2}$ we impose that $G$ equals the Green's function of the

reservoirs

$$G\left(x = -\frac{L}{2}\right) = \frac{1}{2}(\mathbf{1} + \boldsymbol{m}_1 \cdot \boldsymbol{\sigma}) \otimes \frac{1}{\sqrt{(\omega + ih)^2 + |\Delta|^2}} \begin{bmatrix} \omega + ih & |\Delta|e^{-\frac{i\varphi}{2}} \\ |\Delta|e^{\frac{i\varphi}{2}} & -(\omega + ih) \end{bmatrix}$$

$$+ \frac{1}{2}(\mathbf{1} - \boldsymbol{m}_1 \cdot \boldsymbol{\sigma}) \otimes \frac{1}{\sqrt{(\omega - ih)^2 + |\Delta|^2}} \begin{bmatrix} \omega - ih & e^{-\frac{i\varphi}{2}}|\Delta| \\ e^{\frac{i\varphi}{2}}|\Delta| & -(\omega - ih) \end{bmatrix}, \tag{C.1}$$

$$G\left(x = \frac{L}{2}\right) = \frac{1}{2}(\mathbf{1} + \boldsymbol{m}_2 \cdot \boldsymbol{\sigma}) \otimes \frac{1}{\sqrt{(\omega + ih)^2 + |\Delta|^2}} \begin{bmatrix} \omega + ih & |\Delta|e^{\frac{i\varphi}{2}} \\ |\Delta|e^{-\frac{i\varphi}{2}} & -(\omega + ih) \end{bmatrix}$$

$$+ \frac{1}{2}(\mathbf{1} - \boldsymbol{m}_2 \cdot \boldsymbol{\sigma}) \otimes \frac{1}{\sqrt{(\omega - ih)^2 + |\Delta|^2}} \begin{bmatrix} \omega - ih & |\Delta|e^{\frac{i\varphi}{2}} \\ |\Delta|e^{-\frac{i\varphi}{2}} & -(\omega - ih) \end{bmatrix}, \tag{C.2}$$

where $h$ is the effective exchange field induced by the outer ferromagnets. In the middle ferromagnet we obtain the following equations in case the pair amplitudes are small in magnitude:

$$D(1 \pm iP\chi\,\mathrm{sign}(\omega))\partial_{xx}f_{\pm} = 2(|\omega| \pm ih\,\mathrm{sign}(\omega))f_{\pm}, \tag{C.3}$$

$$D(1 + P\gamma)\partial_{xx}f_{\uparrow\uparrow} = 2|\omega|f_{\uparrow\uparrow}, \tag{C.4}$$

$$D(1 - P\gamma)\partial_{xx}f_{\downarrow\downarrow} = 2|\omega|f_{\downarrow\downarrow}, \tag{C.5}$$

while the equations for $\tilde{f}_{\pm}$ and $\tilde{f}$ take exactly the same form. Here, $h$ is the exchange field strength in the middle ferromagnet. If $P$ is considerably large, the contribution of $f_{\pm}$ and $f_{\downarrow\downarrow}$ to the current is negligible and consequently we may only keep $f_{\uparrow\uparrow}$.

From Eqs. (C.1) we may compute $f_{\uparrow\uparrow}$ and $\tilde{f}_{\uparrow\uparrow}$ via

$$f_{\uparrow\uparrow}\left(x = \pm\frac{L}{2}\right) = \frac{1}{4}\mathrm{Tr}\left((\tau_1 + i\tau_2)(\sigma_x + i\sigma_y)G\left(x = \pm\frac{L}{2}\right)\right), \tag{C.6}$$

$$\tilde{f}_{\uparrow\uparrow}\left(x = \pm\frac{L}{2}\right) = \frac{1}{4}\mathrm{Tr}\left((\tau_1 - \tau_2)(\sigma_x - i\sigma_y)G\left(x = \pm\frac{L}{2}\right)\right). \tag{C.7}$$

We may write the solutions to the boundary value problem posed by Eqs. (C.3-C.7) as

$$f_{\uparrow\uparrow}(x) = \frac{i}{\sinh\frac{L}{2\xi}}\,\mathrm{Im}\,\frac{|\Delta|}{\sqrt{(\omega + ih)^2 + |\Delta|^2}}\left(\left(-(m_{2,x} + im_{2,y})e^{i\frac{\varphi}{2}}e^{\frac{L}{2\xi}} - (m_{1,x} + im_{1,y})e^{-i\frac{\varphi}{2}}e^{-\frac{L}{2\xi}}\right)e^{\frac{x}{\xi}}\right.$$

$$\left. + \left((m_{1,x} + m_{2,x})e^{-i\frac{\varphi}{2}}e^{\frac{L}{2\xi}} + (m_{2,x} + im_{2,y})e^{i\frac{\varphi}{2}}e^{-\frac{L}{2\xi}}\right)e^{-\frac{x}{\xi}}\right), \tag{C.8}$$

$$\tilde{f}_{\uparrow\uparrow}(x) = \frac{i}{\sinh\frac{L}{2\xi}}\,\mathrm{Im}\,\frac{|\Delta|}{\sqrt{(\omega + ih)^2 + |\Delta|^2}}\left(\left((-m_{2,x} + im_{2,y})e^{-i\frac{\varphi}{2}}e^{\frac{L}{2\xi}} - (m_{1,x} - im_{1,y})e^{i\frac{\varphi}{2}}e^{-\frac{L}{2\xi}}\right)e^{\frac{x}{\xi}}\right.$$

$$\left. + \left((m_{1,x} - im_{1,y})e^{i\frac{\varphi}{2}}e^{\frac{L}{2\xi}} + (m_{2,x} - im_{2,y})e^{-i\frac{\varphi}{2}}e^{-\frac{L}{2\xi}}\right)e^{-\frac{x}{\xi}}\right), \tag{C.9}$$

where $\xi = \sqrt{\frac{D(1 + \gamma P)}{2|\omega|}}$ is the superconducting coherence length in the ferromagnet for $\uparrow\uparrow$ pairs. The current is given by

$$I = -\sigma_{\uparrow}i(f_{\uparrow\uparrow}\partial_x\tilde{f}_{\uparrow\uparrow} - \tilde{f}_{\uparrow\uparrow}\partial_x f_{\uparrow\uparrow})$$

$$= \frac{2\sigma_{\uparrow}}{\xi\sinh\frac{L}{2\xi}}\left(\mathrm{Im}\,\frac{|\Delta|}{\sqrt{(\omega + ih)^2 + |\Delta|^2}}\right)^2$$

$$\times \left((m_{1,x}m_{2,x} + m_{1,y}m_{2,y})\sin\varphi + (m_{1,x}m_{2,y} - m_{2,x}m_{1,y})\cos\varphi\right). \tag{C.10}$$

Thus, we find that the anomalous current is proportional to $m_{1,x}m_{2,y} - m_{2,y}m_{2,x} = (\boldsymbol{m}_1 \times \boldsymbol{m}_2)_z$. For a generic choice of axis, the anomalous current is proportional to $\boldsymbol{m}_3 \cdot (\boldsymbol{m}_1 \times \boldsymbol{m}_2)$, which confirms Eq. (57) in Sec. 4 in the main text. The usual $\sin\varphi$ contribution of the current is given by $\boldsymbol{m}_1 \cdot \boldsymbol{m}_2 - (\boldsymbol{m}_1 \cdot \boldsymbol{m}_2)_z = (\boldsymbol{m}_1 \times \boldsymbol{m}_3) \cdot (\boldsymbol{m}_2 \times \boldsymbol{m}_3)$.

# D    Explanation of table 1: Structure of tensors in altermagnets

In this section we derive the form that the tensors $T_{jk}$ and $K_{jk}$ may take in the different altermagnetic spin space groups, which are enumerated in Table 1 in the main text, and show that they are only allowed to be nonzero in $d$ - wave altermagnets. Since $T_{jk}$ and $K_{jk}$ obey the same symmetries, we discuss the allowed terms in $T_{jk}$ and keep in mind that the same terms are allowed in $K_{jk}$.

   We encounter two types of symmetries in the material. If the real space operation is not accompanied by a spin flip, the tensor $T_{jk}$ is required to be invariant under this operation. On the other hand, if the real space operation is accompanied by a spin flip, $T_{jk}$ needs to change sign under the real space operation. We use the notation used in [5] for the symmetry elements characterizing the altermagnetic symmetry groups.

## D.1    $d$ - wave altermagnets

There exist 4 collinear spin space groups with a $d$ - wave altermagnet order parameter. The first is $^2m^2m^1m$, which contains 3 independent mirror planes, the first two two of which need to be accompanied by a spin flip. Thus, upon setting $x \to -x$ or $y \to y$, all elements of $T_{jk}$ should change sign, while upon setting $z$ the tensor should remain invariant. There is only one independent term that satisfies these requirements, $T_{xy}$. We conclude the tensor is allowed and takes the form $T = T_{xy} \begin{bmatrix} 0 & 1 & 0 \\ 1 & 0 & 0 \\ 0 & 0 & 0 \end{bmatrix}$.

   The second is $^24/^1m$, that is, there is fourfold rotational symmetry around the z-axis which is accompanied by a spin flip, and a mirror symmetry in the xy-plane without spin flip. The latter disallows $T_{xz}$ and $T_{yz}$, since these change sign under this mirror operation. Since $T_{zz}$ is invariant under rotations around the z-axis, $^24$ disallows this term. For the in-plane components this symmetry implies $T_{xx} = -T_{yy}$. Thus, the allowed terms in $T$ are $T = \begin{bmatrix} T_{xx} & T_{xy} & 0 \\ T_{xy} & -T_{xx} & 0 \\ 0 & 0 & 0 \end{bmatrix}$.

   The third $d$ - wave altermagnetic group is $^24/^1m^2m^1m$. The first two symmetries that characterize this symmetry group are the same symmetries that appeared in the previous group, which leaves $T_{xx}$ and $T_{xy}$ as the only possible independent components. However, this groups contains two extra symmetries. The second mirror operation (yz-plane), which comes along with a spin flip, implies that the tensor changes sign under $x \to -x$, and hence $T_{xx} = 0$. The third symmetry operation represents an independent mirror plane, that is, a mirror plane that can not be obtained by the combination of any of the other symmetry operations. This mirror plane is the (x+y)z-plane, which interchanges $x, y$. Since $T_{xy}$ is indeed invariant under this operation, it is allowed and $T = T_{xy} \begin{bmatrix} 0 & 1 & 0 \\ 1 & 0 & 0 \\ 0 & 0 & 0 \end{bmatrix}$.[2]

---

[2]By defining axes differently, one may also choose the (x+y)z-plane as the second mirror plane, which is accompanied by a spin flip and the yz-plane as the third mirror plane, which is not accompanied by a spin flip. In this case $T_{xx} = -T_{yy}$ are the only possible nonzero elements.

Next we consider the fourth $d$ - wave altermagnet group, $^2 2/^2 m$. The first symmetry operation is a two-fold rotation around the z-axis accompanied by a spin-flip. This means that by setting $x \to -x$ and $y \to -y$ simultaneously, the tensor should change sign. This leaves $T_{xz}$ and $T_{yz}$ as the only possibilities. The second symmetry, a mirror in the xy-plane accompanied by a spin-flip indicates that the tensor should change sign upon sending $z \to z$. This requirement is satisfied by both components. Thus, there are two independent components and $T = \begin{bmatrix} 0 & 0 & T_{xz} \\ 0 & 0 & T_{yz} \\ T_{xz} & T_{yz} & 0 \end{bmatrix}$.

Summarizing, in all of the $d$ - wave altermagnets, the tensor can be nonzero, but which elements are nonzero depends on the symmetry of the specific group. In fact, in two of the groups, $^2 m^2 m^1 m$ and $^2 4/^1 m^2 m^1 m$, there is only one independent nonzero element, and consequently, in those groups $T_{jk}$ and $K_{jk}$ differ only by a constant. On the other hand, in the other two groups, $^2 4/^1 m$ and $^2 2/^2 m$, there are two independent nonzero components, and consequently, $T_{jk}$ and $K_{jk}$ are not related via constant. The results for $d$ - wave superconductors are summarized in the first rows in Table 1.

## D.2  $g$ - wave altermagnets

Next, we consider the $g$ - wave altermagnets. There exist 4 collinear spin space groups that allow for $g$ - wave altermagnetism.

The first group is $^1 4/^1 m^2 m^2 m$. The fourfold rotational symmetry around the z-axis, combined with the identity operator in spin space, implies that $T_{xy} = T_{xz} = T_{yz} = 0$, and $T_{xx} = T_{yy}$. This leaves two independent components, $T_{xx}$ and $T_{zz}$ that are possibly nonzero. The first mirror symmetry, which implies invariance of $T$ under $z \to -z$, does not affect either of them, but the second mirror symmetry implies the tensor should change sign under $x \to -x$, which is not satisfied by either of these terms. Thus $T = 0$.

The second group is $^1 \bar{3}^2 m$. The threefold rotational symmetry accompanied by real space inversion implies $T_{xy} = T_{xz} = T_{yz} = 0$ and $T_{xx} = T_{yy}$. Thus, like in the previous case, we need to consider $T_{xx}$ and $T_{zz}$ The mirror symmetry with spin flip is on a mirror parallel to the rotation axis. This implies the tensor should change sign under $x \to -x$, which both disallows $T_{xx}$ and $T_{zz}$. Hence, $T = 0$.

The third group is $^2 6/^2 m$. The sixfold rotation accompanied by spin-flip implies $T_{zz} = 0$ because this term can not change sign under rotations around the z - axis. Next to this, this symmetry implies a threefold rotational symmetry without spin flip around the $z$-axis, which means $T_{xz} = T_{yz} = 0$. Moreover, by applying the rotation with spin flip three times, we get a $\pi$ rotation with spin-flip, which implies that the tensor should change sign under simultaneously setting $x \to -x$ and $y \to -y$. Therefore, $T_{xx} = T_{yy} = T_{xy} = 0$. With this we have exhausted all independent components, and hence $T = 0$ for this group.

The fourth and last group is $^2 6/^2 m^2 m^2 m$. Since this contains the same sixfold rotational symmetry with spin flip as the previous group, we immediately conclude $T = 0$.

In summary, for all $g$ - wave altermagnets $T = 0$, that is, up to the order considered here, $g$ - wave altermagnets have the same transport properties as antiferromagnets, as summarized in Table 1.

## D.3  $i$ - wave altermagnets

Lastly, we consider the $i$ - wave altermagnets. There are two collinear spin space groups in which this type of altermagnetism may appear. The first symmetry group is $^1 6/^1 m^2 m^2 m$. This group is invariant under six-fold rotations around the $z$-axis. In particular, this means invariance under a $\pi$ rotation, which implies $T_{xz} = T_{yz} = 0$. Comparing the transformation of

the tensor under $\pi/3$ and $-\pi/3$ rotations, we conclude that also $T_{xy} = 0$, while $T_{xx} = T_{yy}$. This leaves two independent terms, $T_{xx}$ and $T_{zz}$. Both of these terms are invariant under the first mirror axis, $z \to -z$ as required. However, they do not flip sign under $x \to -x$, while this is a requirement following the second mirror axis. Thus, $T = 0$.

The second group is $^1m^1\bar{3}^2m$, whose basis elements are a threefold rotation combined with inversion, and two independent mirror planes that both contain the rotation axes, and one of which contains a spin-flip. The invariance under the three-fold rotation axis plus inversion implies $T_{xy} = T_{xz} = T_{yz} = 0$, and moreover $T_{xx} = T_{yy}$. This leaves again $T_{yy}$ and $T_{zz}$ as independent components. Both of the mirrors leave $z$ invariant, but one of them ($^2m$) is accompanied with a spin flip and the other ($^1m$) is not, and this implies $T_{zz} = -T_{zz} = 0$. The first mirror leaves $T_{xx}$ invariant, but the second implies that $T$ should change sign upon $y \leftrightarrow -y$, which disallows $T_{yy}$, the last remaining possible term. We conclude $T = 0$, as summarized in the last two rows of Table 1. Thus, for altermagnets that are not of the $d$ - wave type, the dirty limit transport properties can not be distinguished from those of antiferromagnets.

# E Spin-splitter effect in altermagnets: Normal state

In this section, we present the details of the calculation of the normal state spin-splitter effect discussed in Section 5.1. Specifically, we show that in a finite sample the induced spin current following Eq. (66) leads to magnetization at the boundary of the sample. We consider three different geometries. First we discuss a geometry that is infinite in the transverse direction, to show the generation of a spin current from a voltage difference. Next we study a geometry that is infinite in the longitudinal direction, but has boundaries in the transverse boundaries to illustrate how a spin accumulation arises due to the presence of such boundaries. Lastly, we consider the most realistic setup in which the geometry is finite in both directions and show the spin-splitter effect.

Our starting point is the Usadel equation for collinear altermagnets, Eqs. (59-61) in the main text:

$$\partial_k J_k = [g, \hat{\omega}_{t,t'} \tau_3] + \Gamma_{ab}[g, \tau_3 \sigma_a g \tau_3 \sigma_b] + \mathcal{T}, \tag{E.1}$$

where the matrix current and torque of the system are given by

$$J_k = -D\left(g \partial_k g + \frac{1}{4} P_a T_{jk} \{\tau_3 \sigma_a + g \tau_3 \sigma_a g, g \partial_j g\} + \frac{i}{4} P_a K_{jk}[\tau_3 \sigma_a + g \tau_3 \sigma_a g, \partial_j g]\right), \tag{E.2}$$

while the torque becomes

$$\mathcal{T} = \frac{D}{4} P_a T_{jk}[\tau_3 \sigma_a, \partial_j g \partial_k g] + \frac{iD}{4} P_a K_{jk}[\tau_3 \sigma_a, g \partial_j g \partial_k g]. \tag{E.3}$$

In the normal state, the retarded and advanced components are given by $g^R(t,t') = -g^A(t,t') = \tau_3 \delta(t-t')$, while the Keldysh component satisfies $g^K(t,t') = 2\tau_3 F(t,t')$. Here $F(t,t')$ is the distribution function of the system, which contains $\tau_0$ and $\tau_3$ components. It is easy to check that in the normal case both terms in the torque vanish. The Keldysh part of the matrix current can be expressed in terms of the distribution function as

$$J_k = -D(\partial_k F + \frac{1}{2} P_a T_{jk} \tau_3 \{\sigma_a, \partial_j F\} + \frac{i}{2} P_a K_{jk}[\sigma_a, \partial_j F]). \tag{E.4}$$

We choose the collinear axis to be along the z-direction everywhere. In this case $g$ and $J$ necessarily commute with $\sigma_z$, and consequently, the $K_{jk}$ - term drops out of the equation. From the above equation we can compute the charge current

$$j_e = \frac{\pi \nu}{4} \text{Tr}(\tau_3 J_k(t,t)), \tag{E.5}$$

and spin current

$$j_{k,a}^s = \frac{\pi \nu}{4} \text{Tr}\left(\sigma_z J_k(t,t)\right), \tag{E.6}$$

in terms of the chemical potential

$$\mu = \frac{\pi}{8} \text{Tr}\left(g^K(t,t)\right) = \frac{\pi}{8} \text{Tr}(\tau_3 F(t,t)), \tag{E.7}$$

and spin accumulation

$$\mu_z^s = \frac{\pi}{8} \text{Tr}\left(\tau_3 \sigma_z g^K(t,t)\right) = \frac{\pi}{8} \text{Tr}(\sigma_z F(t,t)), \tag{E.8}$$

that are related to the excess charge and spin via $\delta n = 2\nu\mu$ and $\delta S_z = 2\nu\mu_z^s$.

In Eq. (E.1), we assume an isotropic relaxation term, $\Gamma_{ab} = \frac{1}{\tau^s}\delta_{ab}$. A different structure of $\Gamma_{ab}$ modifies the quantitative results, but does not lead to qualitative changes. By setting $T_{xy} = 1$ as the only independent nonzero entry, we find from Eqs.(E.1-E.2):

$$j_x = -\sigma_D(\partial_x \mu + T_{xy} \partial_y \mu_z^s), \tag{E.9}$$

$$j_y = -\sigma_D(\partial_x \mu + T_{xy} \partial_y \mu_z^s), \tag{E.10}$$

$$j_x^s = -\sigma_D(\partial_x \mu_z^s + T_{xy} \partial_y \mu), \tag{E.11}$$

$$j_y^s = -\sigma_D(\partial_y \mu_z^s + T_{xy} \partial_x \mu), \tag{E.12}$$

$$\partial_k j_k = 0, \tag{E.13}$$

$$\partial_k j_k^s = -\frac{1}{\tau^s} \delta S_z, \tag{E.14}$$

where $\sigma_D$ is the Drude conductivity. The above is the complete set of diffusion equations describing charge and spin transport in diffusive antiferromagnets. Combined with appropriate boundary conditions, they can also describe transport in hybrid structures made of an antiferromagnet attached to normal or ferromagnetic injectors and detectors, in any realistic transport experimental setup. In the next subsections, we use them to describe the creation of spin currents and spin accumulations through electric signals in different geometries.

## E.1 Infinite altermagnetic stripe: Transverse voltage

We first consider a 2D geometry that is infinite in the y-direction. We impose a voltage $\pm\frac{V}{2}$ at $x = \pm\frac{L_x}{2}$, while the spin-voltages vanish at those positions. This fixes the boundary conditions $\mu(\pm\frac{L}{2}) = \pm\frac{V}{2}$ and $\mu_z^s(\pm\frac{L}{2}) = 0$. Thus, in this case, the solution to the Usadel equation Eq. (E.14) is

$$\mu = \frac{V}{L_x} x, \tag{E.15}$$

$$\mu_z^s = 0. \tag{E.16}$$

$$\tag{E.17}$$

Thus, the voltage difference creates a charge current via Eq. (E.9) and transverse spin current via the term proportional to $T_{xy}$ in Eq. (E.12):

$$j_x = -\sigma_D \frac{V}{L_x}, \tag{E.18}$$

$$j_y^s = -T_{xy}\sigma_D \frac{V}{L_x}. \tag{E.19}$$

As we show in the next sections, finite boundaries in the y-direction, will generate a spin-accumulation at these boundaries.

### E.2  Infinite altermagnetic stripe: Electric field along the x-axis

Next, we consider another simple setup: An infinite 2D stripe with boundaries at $y = \pm L_y/2$. We assume a constant electric field, along the x-direction. By translational symmetry $\mu_z^s$ does not depend on $x$, while $\mu = E_x x$. Thus, from Eqs. (E.9-E.14) and the fact that no current may leave the system through the edges with vacuum at $y = \pm\frac{L_y}{2}$, we obtain the following boundary value problem for $\mu_z^s$:

$$D\partial_{yy}\mu_z^s = \frac{1}{\tau^s}\mu_z^s, \tag{E.20}$$

$$\partial_y\mu_z^s(y = \pm\frac{L_y}{2}) = -T_{xy}E_x. \tag{E.21}$$

It is straightforward to obtain the solution of this problem. It reads:

$$\mu_z^s = -T_{xy}E_x l^s \frac{\sinh\left(\frac{y}{l^s}\right)}{\cosh\left(\frac{L_y}{2l^s}\right)}, \tag{E.22}$$

where $l^s = \sqrt{D\tau^s}$ is the spin diffusion length. Thus, we see that biasing a current via an electric field leads to the creation of a spin accumulation at the boundaries in the transverse direction.

### E.3  Finite system

We now consider consider a rectangular altermagnet that is finite in both the $x$ and $y$ directions. Combining the boundary conditions of the two previous problems, that is, using the boundary conditions in the x-direction used in Sec. E.1 and the boundary conditions in the y-direction used in Sec. E.2 we see that for the finite system Eqs. (E.9-E.14) become:

$$(\partial_{xx} + \partial_{yy})\mu + 2T_{xy}\partial_x\partial_y\mu_z^s = 0, \tag{E.23}$$

$$(\partial_{xx} + \partial_{yy})\mu_z^s + 2T_{xy}\partial_x\partial_y\mu = \frac{\mu_z^s}{l_s^2}, \tag{E.24}$$

$$\mu(x = 0) = -\frac{V}{2}, \tag{E.25}$$

$$\mu(x = L_x) = \frac{V}{2}, \tag{E.26}$$

$$\mu_z^s(x = 0, x = L_x) = 0, \tag{E.27}$$

$$(\partial_y\mu + T_{xy}\partial_x\mu_z^s)|_{y=\pm\frac{L_y}{2}} = 0, \tag{E.28}$$

$$(\partial_y\mu_z^s + T_{xy}\partial_x\mu)|_{y=\pm\frac{L_y}{2}} = 0. \tag{E.29}$$

Now, suppose that $T_{xy}$ is small. For $T_{xy} = 0$ only $\mu$ is nonzero, with

$$\mu = \frac{Vx}{L_x}. \tag{E.30}$$

Since $\mu^s$ has no zeroth order component, we find that $\mu$ does not have any first order corrections, we may keep only the zeroth order in $T_{xy}$ of $\mu$, and solve for $\mu_z^s$ to first order in $T_{xy}$. The resulting equations for $\mu_z^s$ read, keeping only terms of first order in $T_{xy}$ and taking into account that $\mu$ does not depend on $y$,

$$\partial_{xx}\mu_z^s + \partial_{yy}\mu_z^s = \frac{1}{(l^s)^2}\mu_z^s, \tag{E.31}$$

$$\mu_z^s(x = 0, x = L_x) = 0, \tag{E.32}$$

$$\partial_y\mu_z^s(y = \pm\frac{L_y}{2}) = -\frac{T_{xy}V}{L_x}. \tag{E.33}$$

We may exploit that boundary conditions in the x-directions to write

$$\mu_z^s = \sum_{n=1}^{\infty} A_n(y) \sin\left(\frac{n\pi}{L_x}x\right). \tag{E.34}$$

The coefficients $A_n(yy)$ have to satisfy the bulk equation. Using that

$$\int_0^{L_x} \sin\left(\frac{n\pi}{L_x}x\right)\sin\left(\frac{m\pi}{L_x}x\right) = \frac{L_x}{2}\delta_{nm},$$

we find that $A_n$ has to satisfy

$$\partial_{yy}A_n(y) = \left(\frac{1}{(l^s)^2} + \frac{n^2\pi^2}{L_x^2}\right)A_n(y), \tag{E.35}$$

as can be obtained by multiplying Eq. (E.31) by $\sin\left(\frac{n\pi}{L_x}x\right)$ and then integrating over the x-direction. A similar procedure applied to the boundary conditions in Eq. (E.33) gives the following boundary conditions for $A_n$:

$$\partial_y A_n(y)\big|_{y=\pm\frac{L_y}{2}} = -\frac{2T_{xy}V}{n\pi L_x}(1 - \cos(n\pi)). \tag{E.36}$$

We note that for all even $n = 2m$, the expression on the right hand side vanishes and the solution to the resulting problem is

$$A_{2m}(y) = 0. \tag{E.37}$$

Thus, there is no contribution from even $n$. On the other hand, for odd $n = 2m + 1$, $m \geq 0$, the boundary terms are nonzero and we have

$$A_{2m+1}(y) = -\frac{4T_{xy}V}{(2m+1)\pi\lambda_{2m+1}L_x}\frac{\sinh(\lambda_{2m+1}y)}{\cosh\left(\lambda_{2m+1}\frac{L_y}{2}\right)}, \tag{E.38}$$

$$\lambda_{2m+1}^2 = \frac{1}{(l^s)^2} + \frac{(2m+1)^2\pi^2}{L_x^2}. \tag{E.39}$$

Thus,

$$\mu_z^s = -\frac{4T_{xy}V}{\pi L_x}\sum_{m=0}^{\infty}\frac{1}{(2m+1)\lambda_{2m+1}}\frac{\sinh(\lambda_{2m+1}y)}{\cosh\left(\lambda_{2m+1}\frac{L_y}{2}\right)}\sin\left(\frac{(2m+1)\pi x}{L_x}\right). \tag{E.40}$$

Thus, there is a nonzero spin accumulation with opposite polarizations at both edges, $y = \pm L_y/2$. This is the spin-splitter effect in diffusive altermagnets discussed in Sec. 5.1 in the main text.

## F  Altermagnet with superconducting correlations: Absence of superconducting spin-splitter effect

In this section, we show explicitly that in an superconducting altermagnet the spin-splitter effect vanishes as claimed by symmetry considerations in Eq. (69-72) in Sec. 5.2 in the main text. We show that triplet pairs do accumulate to the edges, but that these are time-reversal symmetric correlations, that do not lead to any spin accumulation. For clarity of presentation, we consider a 2D altermagnet.

## F.1 Altermagnetic stripe with superconducting correlations

We consider an altermagnet stripe, which has a finite size in the y-direction, and an infinite size in the x direction. The stripe is superconducting or proximitized by a bulk superconductor. A supercurrent is flowing in the x-direction, by imposing a phase gradient, that is, $\Delta = |\Delta|e^{-2iqx}$. We assume this phase gradient is uniform and along the x-direction.

Since we focus on equilibrium properties, we will use the Matsubara formulation and set $\hat{\omega}_{t,t'} \to \omega$ and $g(\omega) = \hat{g}\tau_3 + \hat{f}\tau_1$, where $\omega$ is an arbitrary Matsubara frequency.

In the collinear altermagnet the spin-relaxation tensor takes the form in which $\Gamma_{zz} = \frac{1}{\tau^s}$ is the only nonzero component. In this case we find the following Usadel based on Eqs. (59-61) in Sec. 5.2 the main text:

$$\partial_x J_x + \partial_y J_y = \mathcal{T}, \tag{F.1}$$

$$J_x = -Dg\partial_x g + \frac{DT_{xy}}{4}\sigma_z\{\tau_3 + g\tau_3 g, g\partial_y g\} + \frac{iDK_{xy}}{4}\sigma_z[\tau_3 + g\tau_3 g, \partial_y g], \tag{F.2}$$

$$J_y = -Dg\partial_y g + \frac{DT_{xy}}{4}\sigma_z\{\tau_3 + g\tau_3 g, g\partial_x g\} + \frac{iDK_{xy}}{4}\sigma_z[\tau_3 + g\tau_3 g, \partial_x g], \tag{F.3}$$

$$\mathcal{T} = -\frac{1}{\omega^2 + |\Delta|^2}[\omega\tau_3 + |\Delta|\tau_2 e^{\pm\frac{i}{2}qx\tau_3}, g] + \frac{1}{\tau^s}[g, \tau_3\sigma_z g\tau_3\sigma_z]$$
$$+ \frac{DT_{xy}}{4}\sigma_z[\tau_3, \{\partial_x g, \partial_y g\}] + \frac{iDK_{xy}}{4}\sigma_z[\tau_3, g\{\partial_x g, \partial_y g\}], \tag{F.4}$$

$$J_y\left(y = \pm\frac{d_y}{2}\right) = 0. \tag{F.5}$$

Now consider $\tilde{g} = \tau_3 g^T \tau_3$. Denoting $\tilde{J}_{x,y}$ and $K$ as the corresponding torques, we find that $\tilde{J}_{x,y} = -\tau_3 J_{x,y}^T \tau_3$ and $K = -\tau_3 T^T \tau_3$. Thus, the Green's function $\tilde{g}$ satisfies the system of equations

$$\partial_x \tilde{J}_x + \partial_y \tilde{J}_y = \mathcal{T}, \tag{F.6}$$

$$\tilde{J}_x = -D\tilde{g}\partial_x \tilde{g} + \frac{DT_{xy}}{4}\sigma_z\{\tau_3 + \tilde{g}\tau_3\tilde{g}, \tilde{g}\partial_y \tilde{g}\} + \frac{iDK_{xy}}{4}\sigma_z[\tau_3 + \tilde{g}\tau_3\tilde{g}, \partial_y \tilde{g}], \tag{F.7}$$

$$\tilde{J}_y = -D\tilde{g}\partial_y \tilde{g} + \frac{DT_{xy}}{4}\sigma_z\{\tau_3 + \tilde{g}\tau_3\tilde{g}, \tilde{g}\partial_x \tilde{g}\} + \frac{iDK_{xy}}{4}\sigma_z[\tau_3 + \tilde{g}\tau_3\tilde{g}, \partial_x \tilde{g}], \tag{F.8}$$

$$\mathcal{T} = -\frac{1}{\omega^2 + |\Delta|^2}[\omega\tau_3 + |\Delta|\tau_2 e^{-\frac{i}{2}q\tau_3}, \tilde{g}]$$
$$+ \frac{DT_{xy}}{4}\sigma_z[\tau_3, \{\partial_x \tilde{g}, \partial_y \tilde{g}\}] + \frac{iDK_{xy}}{4}\sigma_z[\tau_3, \tilde{g}\partial_x \tilde{g}\partial_y \tilde{g}], \tag{F.9}$$

$$\tilde{J}_y\left(y = \pm\frac{d_y}{2}\right) = 0. \tag{F.10}$$

This is exactly the same system of equations except for the negation of $g$. Thus, we have $g(q) = \tilde{g}(-q) = \tau_3 g^T(q)\tau_3$. Since $g$ additionally commutes with $\sigma_z$, we find that this means

$$M_z(q) = g\mu_b \frac{\pi\nu}{4}\text{tr}\tau_3\sigma_z g(q) = g\mu_b \frac{\pi\nu}{4}\text{tr}\tau_3\sigma_z\tau_3 g^T(q)\tau_3 = g\mu_b \frac{\pi\nu}{4}\text{tr}\tau_3\sigma_z g(-q) = M_z(-q), \tag{F.11}$$

that is, $M_z$ is even in the phase gradient. This shows that there is no spin-splitter effect in the superconducting state, the magnetization is invariant under a change of the current direction.

Moreover, we find that $\bar{g}(x, y, q) = g(-x, -y, -q)$, which implies that

$$M_z(x, y, q) = M_z(-x, -y, q) = M_z(-x-, y, q). \tag{F.12}$$

Since the equations translationally invariant in the current direction, we find that $M_z$ must be so too, and therefore Eq. (F.12) tells us that the magnetization is even in the transverse direction, and therefore there is not even a difference in spin accumulation along the transverse edges that is even in $q$. In Sec. F.3 we show that this latter property does not hold for inhomogeneous phase gradients, by considering a junction geometry. However, first we will show in Sec. F.2 that in the special orientation in which $K_{xy}$ is the only nonzero component of the K-tensor, the magnetization vanishes throughout the material.

## F.2 Transverse orientation

We consider an altermagnet stripe, which has a finite size in the y-direction, and an infinite size in the x direction. The stripe is superconducting or proximitized by a bulk superconductor. A supercurrent is flowing in the x-direction, by imposing a phase gradient, that is, $\Delta = |\Delta|e^{-2iqx}$. We assume this phase gradient is uniform and along the x-direction.

Since we focus on equilibrium properties, we will use the Matsubara formulation and set $\hat{\omega}_{t,t'} \to \omega$ and $g(\omega) = \hat{g}\tau_3 + \hat{f}\tau_1$, where $\omega$ is an arbitrary Matsubara frequency.

We focus on the case in which $\chi_{axy}$ as only independent nonzero component. The phase gradient is most easily taken care of by using a gauge transformation. As a result of this transformation, we need to set $\partial_x \cdot \to \partial_x \cdot -iq[\tau_3, \cdot]$, while in this gauge $\Delta$ and $g(\omega)$ do not depend on $x$. Therefore, for the Green's function in the y-direction we need to solve, taking into account that $\sigma_a$ commutes with $\tau_3$ and the Green's function the following equations that follow by substituting of $\partial_x g \to -iq[\tau_3, g]$ into Eqs. (59-61) in Sec. 5.2 the main text:

$$\partial_y \tilde{\mathcal{J}}_y = \mathcal{T}, \tag{F.13}$$

$$\mathcal{J}_y = -D(g\partial_y g + \frac{q}{2}\chi_{axy}\sigma_a[\tau_3 + g\tau_3 g, [\tau_3, g]]), \tag{F.14}$$

$$\mathcal{T} = [-\omega\tau_3 - |\Delta|\tau_1, g] + \sigma_a \frac{Dq}{4}\chi_{axy}[\tau_3, g\{\partial_y g, [\tau_3, g]\}] - Dq^2[\tau_3, g\tau_3 g], \tag{F.15}$$

where for simplicity of notation we absorbed all terms that do not belong to the current into the torque.

We show that the solution $g$ has to satisfy $g = \sigma_y g^T \sigma_y = \tilde{g}$, and hence, there is no spin accumulation anywhere. We have

$$\sigma_y J_y^T \sigma_y = -D\sigma_y(\partial_y g^T g^T)\sigma_y - \frac{D}{2}\chi_{axy}\sigma_y[[g^T, \tau_3], \tau_3 \sigma_a^T + g^T \tau_3 \sigma_a^T g^T]\sigma_y$$

$$= D\sigma_y g^T \sigma_y \sigma_y \partial_y g \sigma_y - \frac{D}{2}[\tau_3 \sigma_y \sigma_a^T \sigma_y + g^T \tau_3 \sigma_y \sigma_a^T \sigma_y \sigma_y g^T \sigma_y, [\tau_3, \sigma_y g^T \sigma_y]]$$

$$= D\tilde{g}\partial_y \tilde{g} + \frac{D}{2}[\tau_3 \sigma_a + \tilde{g}\tau_3 \sigma_a \tilde{g}, [\tau_3 \sigma_a, \tilde{g}]] = -\tilde{\mathcal{J}}_y, \tag{F.16}$$

where $\tilde{\mathcal{J}}_y$ is obtained by setting $g \to \tilde{g}$ in $\mathcal{J}_y$ and where it was used that $\sigma_a = -\sigma_y \sigma_a^T \sigma_y$. For the torque we obtain

$$\sigma_y \mathcal{T} \sigma_y = \sigma_y[g^T, \omega\tau_3 + |\Delta|\tau_1]\sigma_y + \frac{Dq}{2}\chi_{axy}\sigma_y[\{[g^T, \tau_3], \partial_y g^T\}g^T, \tau_3 \sigma_a^T]\sigma_y$$

$$+ \frac{-iDq}{2}\gamma_{axy}\sigma_y[\{[g^T, \tau_3], \partial_y g^T\}, \tau_3 \sigma_a^T]\sigma_y - Dq^2\sigma_y[g^T \tau_3 g^T, \tau_3]\sigma_y$$

$$= -[\omega\tau_3 + |\Delta|\tau_1, \tilde{g}] + \frac{Dq}{2}\chi_{axy}[\tau_3 \sigma_y \sigma_a^T \sigma_y, \{[\tau_3, \sigma_y g^T \sigma_y], \sigma_y \partial_y g^T \partial_y\}\sigma_y g^T \sigma_y]$$

$$- \frac{iDq}{4}\gamma_{axy}[\tau_3 \sigma_y \sigma_a^T \sigma_y, \{[\tau_3, \sigma_y g^T \sigma_y], \sigma_y \partial_y g^T \partial_y\}\sigma_y \sigma_y] + Dq^2[\tau_3, \tau_3 \tilde{g}\tau_3]$$

$$= -[\omega\tau_3 + |\Delta|\tau_1, \tilde{g}] - \frac{Dq}{2}\chi_{ajk}[\tau_3 \sigma_a, \{[\tau_3, \tilde{g}], \partial_y \tilde{g}\}\tilde{g}]$$

$$+ \frac{iDq}{2}\gamma_{ajk}[\tau_3 \sigma_a, \{[\tau_3, \tilde{g}], \partial_y \tilde{g}\}] + Dq^2[\tau_3, \tau_3 \tilde{g}\tau_3] = -\tilde{\mathcal{T}}, \tag{F.17}$$

where we used that $\tilde{g}$ anticommutes with both $\partial_y \tilde{g}$ and $[\tau_3, \tilde{g}]$ and defined $\tilde{\mathcal{T}}$ by setting $g \to \tilde{g}$ in $\mathcal{T}$.

Since $\tilde{\mathcal{J}}_y = -\sigma_y \mathcal{J}_y^T \sigma_y$ and $\tilde{\mathcal{T}} = -\sigma_y \mathcal{T}^T \sigma_y$, we conclude that $\tilde{g}$ satisfies the same equation as $g$. Hence,

$$g = \tilde{g} = \sigma_y g^T \sigma_y. \tag{F.18}$$

Since the transformation $\sigma_y \cdot^T \sigma_y$ represents a spin-flip, this implies that there is no spin generated in the system, the $\tau_3$ component of $g$ is proportional to the identity matrix in spin space. This shows that unlike SOC, altermagnetism can not be used in superconducting systems, to generate transverse spin accumulations from charge supercurrents. There is also no spin current. Indeed, since $g = \tilde{g}$, $\text{Tr}(\sigma_a g \partial_y g) = 0$, while for any $g$ we have $\text{Tr}(\sigma_a(\sigma_a[\tau_3 + g\tau_3 g, [\tau_3, g]])) = \text{Tr}([\tau_3 + g\tau_3 g, [\tau_3, g]]) = 0$ because it is a commutator.

Now we show that even if spin-accumulation cannot be induced by supercurrenst in an altermagnet, triplet pairs do exist. For this it is convenient to work with the linearized Usadel equation. For small pair amplitudes we have

$$g = \tau_3 + \begin{bmatrix} 0 & f \\ \bar{f} & 0 \end{bmatrix}, \tag{F.19}$$

where $f$ and $\bar{f}$ are matrices in spin space that can be parameterized as

$$f = \begin{bmatrix} f_+ & 0 \\ 0 & f_- \end{bmatrix}, \tag{F.20}$$

$$\bar{f} = \begin{bmatrix} \bar{f}_+ & 0 \\ 0 & \bar{f}_- \end{bmatrix}. \tag{F.21}$$

Substituting this parameterization in Eqs. (F.13-F.15), we find the following equations for the scalar pair amplitudes $f_\pm$:

$$D\partial_{yy}f_\pm \pm 2DqK_{yy}\text{sign}(\omega)\partial_y f_\pm = 2|\omega|f_\pm - 2|\Delta|, \tag{F.22}$$

$$D\partial_y f_\pm \pm 2DqK_{yy}\text{sign}(\omega)f_\pm\big|_{|y|=\frac{L_y}{2}} = 0, \tag{F.23}$$

while $\bar{f}_\pm$ satisfy

$$D\partial_{yy}\bar{f}_\pm \mp 2DqK_{yy}\text{sign}(\omega)\partial_y \bar{f}_\pm = 2|\omega|\bar{f}_\pm - 2|\Delta|, \tag{F.24}$$

$$D\partial_y \bar{F}_\pm \mp 2DqK_{yy}\text{sign}(\omega)\bar{f}_\pm\big|_{|y|=\frac{L_y}{2}} = 0. \tag{F.25}$$

There are two main conclusions that we may draw from this. First of all, $f_\pm = \bar{f}_\mp$, in line with the symmetry arguments laid out above, and showing that spin is absent. Secondly, $f_+ \neq f_-$, that is, triplets are induced.

The induced triplets are odd-frequency, as expected from triplet s-wave correlations. The solution of this equation is, keeping only terms linear in $K_{yy}$ for simplicity

$$f_\pm = \frac{|\Delta|}{|\omega|} + B_\pm \sinh(\lambda y), \tag{F.26}$$

$$\bar{f}_\pm = \frac{|\Delta|}{|\omega|} + B_\mp \sinh(\lambda y), \tag{F.27}$$

$$\lambda^2 = \frac{1}{D}\left(2|\omega| - Dq^2 K_{yy}^2\right) \approx \frac{2|\omega|}{D}, \tag{F.28}$$

$$B_\pm = \pm\frac{2qK_{yy}}{\lambda}\frac{|\Delta|}{\omega}. \tag{F.29}$$

This shows that to first order in $K_{yy}$ there are indeed odd-frequency triplet correlations that have different sign near the top and the bottom of the material. However, these triplets do not lead to spin accumulation because they satisfy $f_\pm = \bar{f}_\mp$.

## F.3 Junction geometry

The above results are seemingly in contradiction with the results of Ref. [30], in which the authors claimed they one induces a spin accumulation at the transverse edges of a Josephson junction made by an altermagnet and two superconducting electrodes, that is, a 2D S / AM / S junction, by passing a supercurrent. We again describe the same orienation of the altermagnet, that is, the normal of the interface corresponds to a node direction of the altermagnet and $T_{xy}$ and $K_{xy}$ are the only nonzero components. The altermagnet has length $L_x$ and width $L_y$. At $y = \pm\frac{L_y}{2}$ the altermagnet has boundaries with vacuum, while in the superconductors are of the s-wave type and appear for $|x| \geq \frac{L_y}{2}$ and act as electrodes. We use the Kuprianov-Luckichev boundary condition [92] at the boundary between the altermagnets and the superconductors. They contain the boundary parameter $\gamma_B$, the ratio between the boundary conductance and the conductance of the altermagnet in the normal state, which is assumed to be the same for the two boundaries. The superconductors have a pair potential with magnitude $|\Delta|$ and phase $\pm\varphi/2$. In the collinear altermagnet the spin-relaxation tensor takes the form in which $\Gamma_{zz} = \frac{1}{\tau^s}$ is the only nonzero component. In this case we find the following boundary value problem based on Eqs. (59-61) in Sec. 5.2 the main text:

$$\partial_x J_x + \partial_y J_y = \mathcal{T}, \tag{F.30}$$

$$J_x = -Dg\partial_x g + \frac{DT_{xy}}{4}\sigma_z\{\tau_3 + g\tau_3 g, g\partial_y g\} + \frac{iDK_{xy}}{4}\sigma_z[\tau_3 + g\tau_3 g, \partial_y g], \tag{F.31}$$

$$J_y = -Dg\partial_y g + \frac{DT_{xy}}{4}\sigma_z\{\tau_3 + g\tau_3 g, g\partial_x g\} + \frac{iDK_{xy}}{4}\sigma_z[\tau_3 + g\tau_3 g, \partial_x g], \tag{F.32}$$

$$\mathcal{T} = [g, \hat{\omega}_{t,t'}\tau_3] + \frac{1}{\tau^s}[g, \tau_3\sigma_z g\tau_3\sigma_z]$$
$$+ \frac{DT_{xy}}{4}\sigma_z[\tau_3, \{\partial_x g, \partial_y g\}] + \frac{iDK_{xy}}{4}\sigma_z[\tau_3, g\{\partial_x g, \partial_y g\}], \tag{F.33}$$

$$J_y\left(y = \pm\frac{L_y}{2}\right) = 0, \tag{F.34}$$

$$J_x\left(x = \pm\frac{L_x}{2}\right) = \pm\frac{\gamma_B}{\omega^2 + |\Delta|^2}[\omega\tau_3 + |\Delta|\tau_2 e^{\pm\frac{i}{2}\varphi\tau_3}, g]. \tag{F.35}$$

Now consider $\tilde{g} = \tau_3 g^T \tau_3$. Denoting $\tilde{J}_{x,y}$ and $K$ as the corresponding torques, we find that $\tilde{J}_{x,y} = -\tau_3 J_{x,y}^T \tau_3$ and $K = -\tau_3 T^T \tau_3$. Thus, the Green's function $\tilde{g}$ satisfies the system of equations

$$\partial_x \tilde{J}_x + \partial_y \tilde{J}_y = \mathcal{T}, \tag{F.36}$$

$$J_x = -D\tilde{g}\partial_x \tilde{g} + \frac{DT_{xy}}{4}\sigma_z\{\tau_3 + \tilde{g}\tau_3\tilde{g}, \tilde{g}\partial_y\tilde{g}\} + \frac{iDK_{xy}}{4}\sigma_z[\tau_3 + \tilde{g}\tau_3\tilde{g}, \partial_y\tilde{g}], \tag{F.37}$$

$$J_y = -D\tilde{g}\partial_y \tilde{g} + \frac{DT_{xy}}{4}\sigma_z\{\tau_3 + \tilde{g}\tau_3\tilde{g}, \tilde{g}\partial_x\tilde{g}\} + \frac{iDK_{xy}}{4}\sigma_z[\tau_3 + \tilde{g}\tau_3\tilde{g}, \partial_x\tilde{g}], \tag{F.38}$$

$$\mathcal{T} = [\tilde{g}, \omega\tau_3] + \frac{DT_{xy}}{4}\sigma_z[\tau_3, \{\partial_x\tilde{g}, \partial_y\tilde{g}\}] + \frac{iDK_{xy}}{4}\sigma_z[\tau_3, \tilde{g}\partial_x\tilde{g}\partial_y\tilde{g}], \tag{F.39}$$

$$J_y\left(y = \pm\frac{L_y}{2}\right) = 0, \tag{F.40}$$

$$J_x\left(x = \pm\frac{L_x}{2}\right) = \pm\frac{\gamma_B}{\omega^2 + |\Delta|^2}[\omega\tau_3 + |\Delta|\tau_2 e^{\mp\frac{i}{2}\varphi\tau_3}, \tilde{g}]. \tag{F.41}$$

This is exactly the same system of equations. Thus, we have $g(\varphi) = \tilde{g}(-\varphi) = \tau_3 g^T(-\varphi)\tau_3$. Since $g$ additionally commutes with $\sigma_z$, this means that there can not exist any spin accumulation that is odd in the phase difference. Thus, there is no spin-splitter effect, in contrast with example *1a)* of [30], which considered the linearized Usadel equation in an electron gas based altermagnets. We note that this consideration only shows there is no magnetization that is odd in the phase, i.e. odd in the current. Indeed, following Eq. (73), in the junction there might exist a proximity induced magnetization, which is independent of $\varphi$ or even in $\varphi$.

# G  Altermagnet- superconductor hybrid structures: The proximity induced magnetization

In this section we consider a bilayer between a superconductor and a altermagnet, that is, an S / AM junction. We show that magnetization arise in the hybrid structure, even though both materials that constitute the junction do not have any magnetization by themselves. These results confirm the situation described in Fig. 2 in Sec. 5.2 of the main text.

We solve the Usadel linearized equation, that is, like in Sec. 5.2 of the main text, we parameterize

$$g^R = \begin{bmatrix} \text{sign}(\omega) & \hat{f} \\ \hat{\bar{f}} & -\text{sign}(\omega) \end{bmatrix}, \tag{G.1}$$

where the real part of $\hat{g}$ describes the density of states of the material and $\hat{f}, \hat{\bar{f}}$ are the pair amplitudes. Here $f$ is a 4×4 matrix in Nambu-spin space given by

$$f = \begin{bmatrix} 0 & \hat{f} \\ \hat{\bar{f}} & 0 \end{bmatrix}, \tag{G.2}$$

and $\hat{f}, \hat{\bar{f}}$ are 2×2 matrices in spin space. We choose the collinear axis to be along the $z$-direction. In that case it is convenient to write the pair amplitudes as

$$\hat{f} = \begin{bmatrix} f_+ & f_{\uparrow\uparrow} \\ f_{\downarrow\downarrow} & f_- \end{bmatrix}. \tag{G.3}$$

We substitute these expressions in Eqs. (59-61) in Sec. 5.2. After linearization we obtain equations for the anomalous matrices $\hat{f}$ and $\hat{\bar{f}}$ which we solve in three different situations corresponding to the panels a-c of Fig. 2.

## G.1  Longitudinal effect

First we consider a 2D junction between a superconductor ($x < 0$) and altermagnet ($x > 0$). The junction is infinite in the y direction. We assume that the lobe of the altermagnet is along the normal of the interface. In Fig. 2 this corresponds to panel a. In that case, by reflection symmetry in the y-direction $K_{xy} = 0$, but $K_{xx}$ is allowed to be nonzero. We assume that the inverse proximity effect can be neglected and consider the superconducting proximity effect in the altermagnet. The Usadel equation, supplemented with the Kuprianov-Lukichev boundary conditions [92] reads in terms of $f_\pm$:

$$D(1 \pm i\,\text{sign}(\omega)K_{xx})\partial_{xx}f_\pm = 2|\omega|f_\pm, \tag{G.4}$$

$$D(1 \pm i\,\text{sign}(\omega)K_{xx})\partial_x f_\pm(x=0) = -\gamma_B \frac{|\Delta|}{\omega}, \tag{G.5}$$

where $\gamma_B$ is the Kuprianov-Lukichev parameter [92]. The equations for $\tilde{f}_\pm$ are the same and therefore $\tilde{f}_\pm = f_\pm$. The solutions to the boundary value problems are

$$f_\pm = \gamma_B \frac{|\Delta|}{\omega} \frac{\tilde{\xi}}{1 \pm i\mathrm{sign}(\omega)K_{xx}} e^{-\frac{x}{\tilde{\xi}}}, \tag{G.6}$$

$$\tilde{\xi} = \sqrt{\frac{D(1 \pm iK_{xx})}{2|\omega|}}, \tag{G.7}$$

and $\tilde{f}_\pm = f_\pm$. Thus, to second order in the pair amplitudes, the magnetization is given by

$$M_z = -ig\mu_B \frac{\pi\nu}{4}\mathrm{Tr}(\tau_3\sigma_z g) \approx 2g\mu_B \frac{\pi\nu}{4}(f_-^2 - f_+^2)$$

$$= -2\pi TK_{xx}\sum_\omega 2g\mu_B \frac{\pi\nu}{4}\gamma_B^2 \frac{|\Delta|^2}{\omega^2}\mathrm{Re}\frac{\tilde{\xi}}{1+iK_{xx}}e^{-\frac{x}{\tilde{\xi}}}\mathrm{Im}\frac{\tilde{\xi}}{1+iK_{xx}}e^{-\frac{x}{\tilde{\xi}}}. \tag{G.8}$$

For any nonzero $K_{xy}$ this expression is finite, which shows that the hybrid structure of two materials without magnetization, a superconductor and an altermagnet, yields a finite magnetic moment at the boundary between the two materials.

To first order in $K_{xy}$ this expression may be written as

$$M_z(x) \approx 2g\mu_B K_{xy}\frac{\pi\nu}{4}\gamma_B^2 \frac{|\Delta|^2}{(\pi T)^2}D\sum_n \frac{1}{(2n+1)^3}e^{-\sqrt{\frac{2\pi T(2n+1)}{D}}2x}. \tag{G.9}$$

Thus, at the boundary, $x = 0$ this reads, using that $\sum_n(2n+1)^{-3} = \frac{7\zeta(3)}{8}$:

$$M_z(x=0) \approx \frac{7\zeta(3)}{4}g\mu_B \frac{\pi\nu}{4}\gamma_B^2 \frac{|\Delta|^2}{(\pi T)^3}D. \tag{G.10}$$

Meanwhile, for $x \gg \xi_T = \sqrt{\frac{D}{\pi T}}$ we have

$$M_z(x \gg \xi_T) \approx 2g\mu_B \frac{\pi\nu}{4}\gamma_B^2 \frac{|\Delta|^2}{(\pi T)^2}De^{-\frac{x}{\xi_T}}, \tag{G.11}$$

which shows that $M_z$ decays over a distance $\xi_T$. This effect is illustrated in Fig. 2a in Sec. 5.2 in the main text.

## G.2 Transverse effect

Next, we consider a similar setup, but now the lobes of the altermagnet are oriented in the transverse direction. In this case the reflection $y \to -y$ changes the orientation of all spins, and therefore $K_{xx} = 0$ but $K_{xy} \neq 0$. This happens if the normal to the interface corresponds to a node of the altermagnet, and consequently the system is invariant under a mirror in the y-direction accompanied by a spin-flip. In Fig. 2 in the main text this corresponds to panel b. If the geometry is infinite in the y-direction, the pair amplitudes do not depend on the y-coordinate and consequently, $K_{jk}$ does not have any influence on the proximity effect, and consequently there is no spin. This however changes, if we introduce boundaries in the transverse y-direction. To this end, we consider a finite junction, with length $L_x$ in the x-direction and width $L_y$ in the y-direction.

We consider the following boundary value problem in $f_s, f_t$ such that $\hat{f} = f_s + f_t$

$$D(\partial_{xx} + \partial_{yy})f_s + 2iD\mathrm{sign}(\omega)K_{xy}\partial_x\partial_y f_t = 2|\omega|f_s, \tag{G.12}$$

$$D(\partial_{xx} + \partial_{yy})f_t + 2iD\mathrm{sign}(\omega)K_{xy}\partial_x\partial_y f_s = 2|\omega|f_t, \tag{G.13}$$

with boundary conditions

$$D\partial_x f_s + iD\text{sign}(\omega)K_{xy}\partial_y f_t = -2\gamma_B D\frac{|\Delta|}{\omega}, \qquad x = -\frac{L_x}{2}, \tag{G.14}$$

$$D\partial_x f_s + iD\text{sign}(\omega)K_{xy}\partial_y f_t = 0, \qquad x = \frac{L_x}{2}, \tag{G.15}$$

$$D\partial_x f_t + iD\text{sign}(\omega)K_{xy}\partial_y f_s = 0, \qquad x = \pm\frac{L_x}{2}, \tag{G.16}$$

$$D\partial_y f_s + iD\text{sign}(\omega)K_{xy}\partial_x f_t = 0, \qquad y = \pm\frac{L_y}{2}, \tag{G.17}$$

$$D\partial_y f_t + iD\text{sign}(\omega)K_{xy}\partial_x f_s = 0, \qquad y = \pm\frac{L_y}{2}. \tag{G.18}$$

We assume that $K_{xy}$ is small and expand in this quantity. To zeroth order in $K_{xy}$ we have $f_t = 0$. The solution is the usual solution for the proximity effect of a superconductor on a normal metal:

$$f_s = 2\gamma_B \xi \frac{|\Delta|}{|\omega|} \frac{\cosh\frac{x-\frac{L_x}{2}}{\xi}}{\sinh\frac{L_x}{\xi}}, \tag{G.19}$$

$$\xi^2 = \frac{D}{2|\omega|}. \tag{G.20}$$

Now, to first order in $K_{xy}$ we obtain the following boundary value problem for $f_t$, keeping in mind that $f_s$ only depends on $x$, not on $y$:

$$D(\partial_{xx} + \partial_{yy})f_t = 2|\omega|f_t, \tag{G.21}$$

$$\partial_x f_t = 0, \qquad x = \pm\frac{L_x}{2}, \tag{G.22}$$

$$\partial_y f_t = -iK_{xy}\text{sign}(\omega)\partial_x f_s = -2iK_{xy}\text{sign}(\omega)\gamma_B \frac{|\Delta|}{|\omega|} \frac{\sinh\frac{x-\frac{L_x}{2}}{\xi}}{\sinh\frac{L_x}{\xi}}, \quad y = \pm\frac{L_y}{2}. \tag{G.23}$$

Because of Eq. (G.22) we may apply a Fourier transformation and write

$$f = C_0(y) + \sum_{m>0} C_m(y)\cos\left(m\pi\frac{x+\frac{L_x}{2}}{L_x}\right). \tag{G.24}$$

For the coefficients $C_m(y)$ we obtain a 1D differential equations, which can be obtained my multiplying Eqs. (G.21-G.23) by $\cos\left(m\pi\frac{x+\frac{L_x}{2}}{L_x}\right)$ and integrating over the x-direction. Specifically, for $C_0$ we get the equation

$$D\partial_{yy}C_0 = 2|\omega|C_0, \tag{G.25}$$

$$\partial_y C_0 = 2iK_{xy}\text{sign}(\omega)\gamma_B \frac{\xi}{L_x}\frac{|\Delta|}{|\omega|}\tanh\frac{L_y}{2\xi}, \qquad y = \pm\frac{L_y}{2}. \tag{G.26}$$

This is a 1D diffusion equation, whose solution by virtue of the boundary conditions is anti-symmetric in $y$:

$$C_0(y) = 2iK_{xy}\text{sign}(\omega)\gamma_B \frac{\xi^2}{L_x}\frac{|\Delta|}{|\omega|}\tanh\frac{L_y}{2\xi}\frac{\sinh\frac{y}{\xi}}{\cosh\frac{L_y}{2\xi}}. \tag{G.27}$$

Meanwhile, for $m > 0$ we have

$$D\partial_{yy}C_m = \left(2|\omega| + m^2\pi^2\frac{D}{L_x^2}\right)C_m, \tag{G.28}$$

$$\partial_y C_m = 4iK_{xy}\text{sign}(\omega)\gamma_B\frac{\xi}{L_x}\frac{|\Delta|}{|\omega|}\frac{L_x^2}{L_x^2+m^2\pi^2\xi^2}\tanh\frac{L_y}{2\xi}, \qquad y = \pm\frac{L_y}{2}. \tag{G.29}$$

Again this is a 1D diffusion equation whose solution needs to change sign upon inversion of $y$. Thus, its solution is

$$C_m(y) = 4iK_{xy}\text{sign}(\omega)\gamma_B\frac{\xi\xi_m}{L_x}\frac{|\Delta|}{|\omega|}\tanh\frac{L_y}{2\xi}\frac{L_x^2}{L_x^2+m^2\pi^2\xi^2}\frac{\sinh\frac{y}{\xi_m}}{\cosh\frac{L_y}{2\xi_m}}\cos\left(m\pi\frac{x+\frac{L_x}{2}}{L_x}\right). \tag{G.30}$$

Inserting Eqs. (G.27) and (G.30) into Eq. (G.24), we find that $f_t$ is given by

$$f_t(x,y) = 2i\,\text{sign}(\omega)\frac{\xi}{L_x}K_{xy}\gamma_B\frac{|\Delta|}{|\omega|L_x} \tag{G.31}$$

$$\times\tanh\frac{L_x}{2\xi}\left(\xi\frac{\sinh\frac{y}{\xi}}{\cosh\frac{L_y}{2\xi}} + 2\sum_m\frac{L_x^2}{L_x^2+m^2\pi^2\xi^2}\xi_m\frac{\sinh\frac{y}{\xi_m}}{\cosh\frac{L_y}{2\xi_m}}\cos\left(m\pi\frac{x+\frac{L_x}{2}}{L_x}\right)\right),$$

$$\xi_m^{-2} = \xi^{-2} + \frac{m^2\pi^2}{L_x^2}. \tag{G.32}$$

The magnetization $M_z$ can then be calculated from this via

$$M_z = 2g\mu_B\frac{\pi\nu}{4}\sum_\omega\text{Im}(f_s f_t),$$

$$= g\mu_B\pi\nu\frac{\xi^2}{L_x}K_{xy}\gamma_B^2\frac{|\Delta|^2}{\omega^2 L_x} \tag{G.33}$$

$$\times\tanh\frac{L_x}{2\xi}\frac{\cosh\frac{x-\frac{L_x}{2}}{\xi}}{\sinh\frac{L_x}{\xi}}\left(\xi\frac{\sinh\frac{y}{\xi}}{\cosh\frac{L_y}{2\xi}} + 2\sum_m\frac{L_x^2}{L_x^2+m^2\pi^2\xi^2}\xi_m\frac{\sinh\frac{y}{\xi_m}}{\cosh\frac{L_y}{2\xi_m}}\cos\left(m\pi\frac{x+\frac{L_x}{2}}{L_x}\right)\right).$$

This is in general nonzero. Numerical evaluation of this expression for $L_x = 2\xi$ and $L_y = \xi$ results in Fig. 5. Our results show that in this orientation there is no magnetization averaged over the entire edge, but instead magnetizations near the corners of the material arise. The effect is schematically illustrated in Fig. 2b in Sec. 5 in the main text.

### G.3   Superconducting island on top of an altermagnet

In the previous subsections we saw that if the normal of the interface corresponds to a lobe direction of the altermagnet, a magnetization is induced. In contrast, if the normal of the interface corresponds to a node direction of the altermagnet, there may only be a transverse effect. Using these results, one can infer what happens if a circular superconducting island is placed on top of an antiferromagnet, similar to the situation in panel c of Fig. 2 in the main text. If the island radius $R$ is much larger than the coherence length $\xi$, the boundary locally resembles that of the extended junctions discussed in the previous subsections. Thus, a finite magnetization will appear in those directions corresponding to the altermagnet lobes, positive along one type of lobes, negative along the other. This occurs at distances of the order $\xi$ around the islands, as discussed in Sec. G.1. On other hand, along node directions, we conclude from Sec. G.2 that the magnetization vanishes. This yields the magnetization pattern discussed in Fig. 2c. A qualitative description of the proximity induced magnetization effect can be obtain by solving Eqs. (59-61) in Sec. 5.2 in a 2D situation.

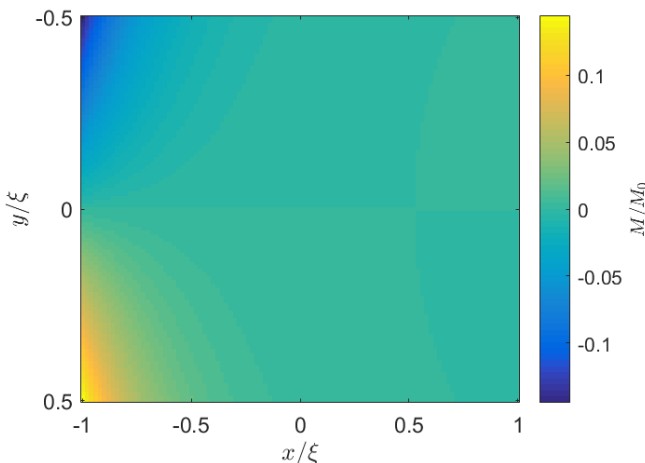

Figure 5: The transverse spin accumulation in a finite size 2D altermagnet with $K_{xy} \neq 0$ and $K_{xx} = 0$, a boundary with a superconductor at $x = -\frac{L_x}{2} = -\xi$ and a boundary with vacuum at $x = \frac{L_x}{2} = \xi$ and $|y| = \frac{L_y}{2} = \frac{\xi}{2}$. We normalized the magnetization by $M_0 = 2g\mu_B = \pi\nu K_{xy}\xi^3\gamma_B^2\frac{|\Delta|^2}{(\pi T)^2 L_x^3}\tanh\frac{L_x}{2\xi}$, where $\xi_0 = \sqrt{\frac{D}{2\pi T}}$.

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
