# Peer review of "Quantum transport theory for unconventional magnets: Interplay of altermagnetism and p-wave magnetism with superconductivity"

_SciPost Physics, doi:SciPost Phys. 18, 178 (2025)_

## Round 1 · Referee Report · Anonymous (Referee 1) · 2025-3-14

Report

In this work, the authors introduce a unified framework to describe the transport of itinerant spin and charge across a magnetic conductor in both normal and superconducting states. This framework is based on the extension of the Keldysh non-linear sigma model (KNLSM) for disordered electrons to magnetic conductors, where the thermal average of the order parameter corresponds to the Green function of the electron fluid. The particular functional form of the action is determined, beyond the KNLSM, from (spin-group) symmetry considerations, and its saddle-point equations represent a generalization of the Usadel equations for the superconducting state, whose validity holds in a wide range of temperatures (far from the critical point). The authors apply their general framework to different classes of magnetic materials, such as ferromagnets and antiferromagnets (conventional magnetism), as well as altermagnets and p-wave magnets (unconventional magnetism), and they predict exciting new physics in both normal and superconducting phases.

The results obtained here are very interesting as well as timely, and the physics/methodology discussed seems overall correct. That being said, I would like to raise the following concerns/comments:

1- In Eq, (3), does the $\tau_{3}$ multiply $\xi_{a}$ instead of $\xi_{s}$? 2- Throughout the text, the authors refer to $\phi$ as the superconducting phase, but in the formulas it reads $\varphi$. 3- In Eq. (22), there is a matrix $\sigma_{a}$ missing in the term parametrized by the tensor $\chi$ 4- Why are Eqs. (23), (59) and (84) different? The superconducting term is either absent or have different forms in the first commutator of the right-hand side of the equation. 5- Does the minus sign in Eq. (32) only affects the first term $g\partial_{k}g$? If this equation is derived from Eq. (24), it should be like this. The same applies to Eqs. (60) and (85). 6- The authors should provide a reference (or a derivation in the appendices) for Eq. (34). 7- I have found a discrepancy between Eqs. (42)-(43) and (45)-(46) if Eq. (44) is correct. Is there a factor 2 missing in the term $\gamma P$ of Eq. (44)? In addition, does $P$ in this equation refer to $P_{z}$? 8- The authors should provide a reference for Eq. (47). 9- In Eq. (48), what is the definition of $\tilde{f}$? I think that it is provided in Eq. (F6), but it should be also given in the main text. 10- a) In Eqs. (53) and (54), should $\tilde{f}$ be replaced by $f$ in the first two terms of the right-hand side? b) Should it read $\partial_{x}\tilde{f}_{\downarrow\downarrow}$ in the second term of the right-hand side?. c) What is the value of/expression for $\sigma$ in these equations? 11- In Eq. (56), it should read $j(-h)^{*}$ instead of $J(-h)^{*}$ 12- According to Eqs. (28) and (29), $D$ should not be a global prefactor in Eq. (60). The same applies to Eq. (61) for the torque and to Eq. (85) for the current. 13- In the subsection ‘Superconductivity and altermagnets’, how are $f_{s}$ and $f_{t}$ related to $f_{0}$ and $f_{z}$ introduced after Eq. (49)?. If they are the same, I would recommend the authors to unify the notation. 14- In Eqs. (67) and (68), there is a $-D$ prefactor missing in front of the $\partial_{x}f$ terms (see comment #12). In addition, which component of the polarization axis does $P^{a}$ refer to? 15- After Eq. (69), what does $q_{k}$ refer to? Is it the superconducting phase gradient $q$? 16- Again, which component does $P^{a}$ refer to in Eq. (72)? The left-hand side of this equation represents a scalar, whereas the right-hand side represents a vector (indexed by $a$). 17- In Eq, (74), what does $\gamma_{B}$ mean? The Kuprianov-Lukichev boundary conditions are formally introduced in Appendix G and no discussion of them is found in the main text. 18- In Eqs. (73)—(80) there are issues with the global prefactor $D$ and the value of $P^{a}$ along the lines of comments #12 and #16 19- In Eq. (87), the prefactor of the second term should read $(1-\sigma_{z})$ 20- In Eq. A15, the third term should contain $\tau_{3}\sigma_{y}\sigma_{a}\sigma_{y}$ instead of $\tau_{3}\sigma_{a}\sigma_{y}\sigma_{a}\sigma_{y}$. The same applies to the third term of Eq. (A19). 21- After Eq. (A20), I guess that the authors mean that the tensors are even/odd in spin indices. 22- In Eq. (B7), there are subindices ‘$j$’ missing in the partial derivatives of $Q$ appearing in the functional derivatives of the action. 23- How are Eqs. (C3)-(C5) related to Eqs. (50)-(52)? For example, there is a prefactor 2 appearing on the right-hand side of Eqs. (C3)-(C5) that is not present in Eqs. (50)-(52). 24- What is the definition of $h_{m}$ and $\tilde{h}$ in Eqs. (C3)-(C5)? 25- After Eq. (C9), is the expression for the superconducting coherence length $\xi$ correct? (for example, $P\longrightarrow\gamma P$?) 26- In Eq. (C10), which is the effect of the conductivities $\sigma_{\uparrow}$, $\sigma_{\downarrow}$ [see Eq. (53)]? 27- Similar to the comment #12, in Eqs. (E1)—(E4) there are issues with the global prefactor $D$ and the use of the tensors $T$ and $K$. 28- Before Eq. (E20) and in Eq. (E21), the boundary condition should be applied at $y=\pm L_{y}/2$, not at $z=\pm L_{y}/2$ 29- In Eq. (E20) should it read $\sigma_{D}$ instead of $D$? 30- The right-hand side of Eq. (E24) should read $(1/l^{s})^{2}\mu_{z}^{s}$ 31- Unify the subindex of $\lambda$ in Eqs. (E38) and (E39). 32- Check the global prefactor $D$ in Eqs. (F10)—(F13) 33- Are there terms missing in Eq. (F27)?

I would recommend the publication of this manuscript in SciPost Physics if the authors address the above concerns satisfactorily.

Recommendation

Ask for major revision

  • validity: top
  • significance: high
  • originality: top
  • clarity: high
  • formatting: excellent
  • grammar: excellent

Author:  Tim Kokkeler  on 2025-04-23  [id 5404]

(in reply to Report 1 on 2025-03-14)
Category:
correction

We thank the Referee for their assessment of our manuscript and the useful comments. We have included our response in the attachment.

Attachment:

Referee_1_response.pdf

---

## Round 1 · Referee Report · Anonymous (Referee 2) · 2025-4-16

Strengths

This paper discusses very current topic about altermagnet / superconductor junctions. They presented a general and reasonable theoretical scheme and predicted very interesting quantum phenomena.

Report

In this paper, the authors have presented the quasiclassical transport theory for materials with magnetic exchange and superconducting correlations. Starting from the Keldysh nonlinear sigma model, they have derived effective low-energy action based on spin space group and transport equations, which are obtained as the saddle point equations of this action. Their theory is very generic and useful for the study of ferromagnet junctions including altermagnet and p-wave magnet. They have predicted spontaneous magnetic moments in superconductor / altermagnet junction and spin-galvanic effect in p-wave magnet junctions. Since studying these unconventional magnets is a current hot topic now, the obtained formulation and results in superconductor / unconventional magnet junction are really timely and exciting. Thus, I appreciate this paper and recommend publication. Before publication, I would like to ask the authors to consider the following points.

1. It is better to explain physically why eq. (10) is obtained.
2. Please check eq. (15) again.
3. I would like to ask about the magnitude of magnetic order. To be consistent with quasiclassical theory, it is much smaller and should be the same order of the pair potential of superconductors. Is it true?
4. The authors have discussed the spin-triplet pairing in various places in superconductor / magnet junctions. Since magnets are diffusive, I think the resulting spin-triplet pairing should be odd-frequency. Is it really true? Since odd-frequency pairing has been studied up to now in superconducting junctions (Refs. Rev. Mod. Phys. 77, 1321, 2005, J. Phys. Soc. Jpn. 81, 011013, 2012). The discussion about the odd-frequency pairing is appreciated.
5. Below eq. (82), it is written that “there is an intermediate angle for which the anomalous current vanishes.” Can you predict the intermediate angle?
6. How about the condition of anomalous current appears? Is it the case both the time reversal symmetry and spatial inversion symmetry are broken?
7. Below eq. (86), there is a discussion about the p-wave magnet with homogeneous superconductor. There is a relevant work in arXiv:2501.08646.

Recommendation

Publish (surpasses expectations and criteria for this Journal; among top 10%)

  • validity: high
  • significance: top
  • originality: top
  • clarity: high
  • formatting: excellent
  • grammar: excellent

Author:  Tim Kokkeler  on 2025-04-23  [id 5405]

(in reply to Report 2 on 2025-04-16)
Category:
correction

We thank the Referee for their assessment of our manuscript and the useful comments. In the attached file we provide our response to the points raised by the Referee.

Attachment:

Referee_2_response.pdf

---

## Round 3 · Referee Report · Anonymous (Referee 1) · 2025-4-28

Report

In my opinion, the authors have addressed the concerns raised by the Referees satisfactorily in the revised version of the manuscript. I would recommend the publication of the present version (with minor corrections) in SciPost Physics.

Some minor issues to be addressed:

1) In Eq. (31), the matrix pair potential $\hat{\Delta}$ is missing in the first commutator. 2) I still think that there is a problem with formula (53). Compare it with the expression used in Eq. (C10) for the calculation of the charge supercurrent. 3) If Eq. (E20) is correct, the authors should then check Eq. (E14).

Recommendation

Publish (surpasses expectations and criteria for this Journal; among top 10%)

---

## Round 3 · Author Response

Dear Editors,

We would like to thank both referees for their careful examinations of the manuscript, their positive assessment of our article, and their suggestions which we could use to improve our manuscript. We have implemented all comments and suggested changes in the updated manuscript.

---

## Round 3 · List of Changes

A point by point overview of the changed is listed in the response to the Referees

---

## Editorial Decision

published